# Vision Transformers with Self-Distilled Registers

**Zipeng Yan**[*1]    **Yinjie Chen**[*1,2]    **Chong Zhou**[3]    **Bo Dai**[1]    **Andrew F. Luo**[†1]

[1] University of Hong Kong    [2] Zhejiang University    [3] Nanyang Technological University

* Equal contribution – Co-First authors

`maxwellcaffrey915@gmail.com, kelvinyzp@gmail.com`    [†]Corresponding author: `aluo@hku.hk`

## Abstract

Vision Transformers (ViTs) have emerged as the dominant architecture for visual processing tasks, demonstrating excellent scalability with increased training data and model size. However, recent work has identified the emergence of artifact tokens in ViTs that are incongruous with local semantics. These anomalous tokens degrade ViT performance in tasks that require fine-grained localization or structural coherence. An effective mitigation of this issue is the addition of register tokens to ViTs, which implicitly "absorb" the artifact term during training. Given the availability of existing large-scale pre-trained ViTs, in this paper we seek to add register tokens to existing models without retraining the models from scratch, which is infeasible considering their size. Specifically, we propose Post Hoc Registers (**PH-Reg**), an efficient self-distillation method that integrates registers into an existing ViT without requiring additional labeled data and full retraining. PH-Reg initializes both teacher and student networks from the same pre-trained ViT. The teacher remains frozen and unmodified, while the student is augmented with randomly initialized register tokens. By applying test-time augmentation to the teacher's inputs, we generate denoised dense embeddings free of artifacts, which are then used to optimize only a small subset of unlocked student weights. We show that our approach can effectively reduce the number of artifact tokens, improving the segmentation and depth prediction of the student ViT under zero-shot and linear probing. Our code is publicly available at this repository.

## 1 Introduction

Vision Transformers (ViTs) are now the dominant architecture in visual modeling, delivering strong performance across classification, detection, and segmentation. Unlike convolutional networks with their built-in locality inductive bias, ViTs process images by spatially splitting them into patches and applying self-attention to enable global feature interactions. This architectural design leads to superior scalability, particularly with contrastive or self-supervised pre-training objectives, and facilitates more flexible representation learning, as it is less constrained by the translation invariance assumptions inherent in CNNs. This flexibility enables remarkable emergent capabilities. Models like CLIP, trained solely on image-text alignment, achieve competitive open-vocabulary segmentation through zero-shot dense queries; while self-supervised approaches learn semantically rich features directly from unlabeled images.

However, the same data-driven attention mechanisms that enable ViT's representation power can also lead to the emergence of *artifact tokens*. These are outlier features often discordant with local image semantics, meaning they fail to correspond to locally meaningful image structures. The propensity for ViTs to generate such tokens is exacerbated by their lack of strong, built-in spatial priors, which can result in inconsistent dense representations. Ultimately, the presence of these artifact tokens

39th Conference on Neural Information Processing Systems (NeurIPS 2025).

| Image | MaskCLIP | SCLIP | NACLIP | Ours (PH-Reg) |
|-------|----------|-------|--------|---------------|

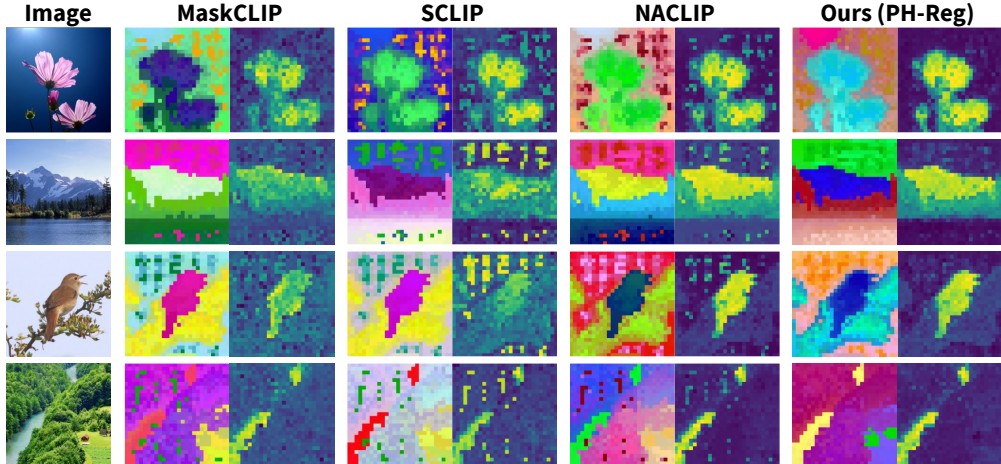

Figure 1: **Effect of PH-Reg on Open-vocabulary Segmentation.** For each image, we compare four methods: MaskCLIP which directly takes the *value* features from the last attention layer; SCLIP which adds correlative self-attention; NACLIP which further enforces a locality bias; and our PH-Reg method with self-distilled registers. We utilize the same `OpenAI CLIP ViT-B/16` weights for all three methods. For each method, we visualize the UMAP of the dense features and a heatmap of one text query. Our method yields noticeably cleaner dense features and high quality localizations, and requires only a small set of additional register parameters compared to the original network.

disrupts fine-grained localization, a critical capability for tasks demanding high spatial precision, such as detailed semantic segmentation or part identification.

Recent work has sought to mitigate artifact tokens via architectural modifications, where *register tokens* are added to the network. These register tokens are randomly initialized, with learnable parameters that participate in the self-attention process similar to the `[CLS]` token, but are not otherwise used during the output. Although these register tokens are not explicitly supervised during training, they effectively "absorb" the artifact term and learn to attend to global objects. While effective, introducing register tokens constitutes a fundamental architectural modification that requires training from scratch—a time-consuming and computationally demanding process. This significantly limits their applicability, especially given the vast ecosystem of *existing, high-performing pre-trained vision models.*

We present a solution to this issue with **P**ost **H**oc **Reg**isters (**PH-Reg**), an efficient self-distillation framework that requires no labeled data or full retraining. We illustrate OpenAI CLIP with PH-Reg in Figure 1. In PH-Reg, both teacher and student networks are initialized from the same pre-trained model weights. And the only extra parameters are the register tokens added to the student network. Our proposed framework freezes the teacher during training. Images provided to the teacher undergo test-time augmentation (e.g. random offsets and horizontal flips). This augmentation strategy effectively denoises the teacher's dense features without requiring gradient-based updates on the teacher itself, yielding stable dense targets. The denoised dense features are used as a distillation target for the student network, where only a small set of parameters are optimized. This entire process requires only a modest set of unlabeled images, enabling significant enhancements to pre-trained models with minimal computational overhead. Concretely our contributions are as follows: **1.** We propose a test-time augmentation scheme that can effectively denoise dense features in vision transformers. Our denoiser does not require costly neural fields and does not require gradient based optimization. **2.** We elucidate the underlying components in a student model that contribute to learning a clean dense feature map. We show that by finetuning select weights, we are able to achieve clean dense features with minimal additional parameters. **3.** We demonstrate that PH-Reg effectively improves the consistency of dense feature representations in ViTs, leading to quantifiable improvements on downstream tasks that rely on fine-grained spatial understanding (e.g., semantic segmentation or depth prediction). Our method preserves the original utility of dense features without inducing unwanted distribution shift, and functions well with zero-shot language-based dense queries.

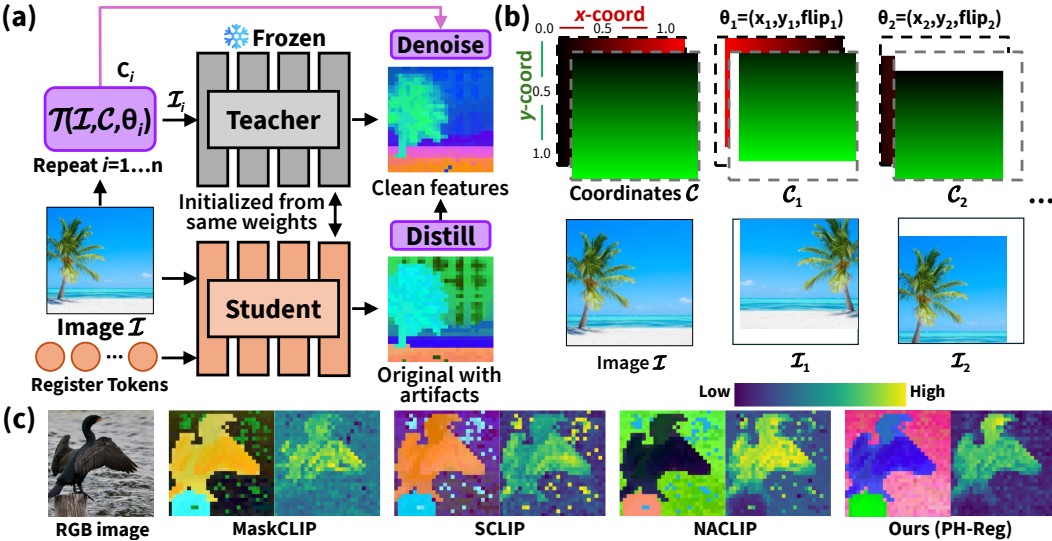

Figure 2: **Learning Framework of PH-Reg. (a)** Our framework begins by creating two networks from the same set of weights. In the teacher, the weights are frozen and unmodified. In the student, the only additional parameters are learnable register tokens. The teacher creates a learning target using denoised representations. **(b)** An image $\mathcal{I}$ undergoes augmentation by function $\mathcal{T}$ with random augmentation parameters consisting of random offsets and horizontal flips.**(c)** Given an RGB image, we utilize UMAP to visualize the features, and a heatmap using CLIP text query. Our method can produce significantly cleaner dense representations with minimal additional inference cost.

## 2 Related Work

**Transformers in Visual Learning.** Building upon the success of self-attention in language modeling, architectures that leverage transformer based token-mixing have been proposed for visual generation [1, 2, 3] and recognition tasks [4, 5], cumulating in the ViT architecture which relies on very few locality biases [6]. In the years since, many improvements and variants have been proposed [7, 8, 9, 10, 11, 12]. The improvements have largely focused on data [13] and compute efficiency [14, 15, 16, 17, 18, 19, 20, 21, 22]. In general, vision transformers tokenize an image into a set of patches, where each patch is first processed using an MLP or convolution block [23, 24, 25, 26], the patches are further processed with self-attention which enables global token interactions beyond those in convolutional networks. As self-attention is permutation invariant, positional information is typically injected using learnable positional embeddings or relative positions [27, 28, 29, 16, 30, 31, 32]. Positional embeddings have been suggested to play a role in the emergence of dense ViT artifacts, as networks with positional embeddings removed have smooth feature maps [33].

**Representation Learning with Vision Transformers.** The lack of restrictive local inductive biases in Vision Transformers enables strong scaling behavior across a diverse set of tasks. Beyond traditional supervised learning on categorical datasets such as ImageNet, methods have been proposed to learn on large scale datasets by leveraging language contrastive objectives [34, 35], or self-supervised image-level objectives [36, 37, 38] and patch-level objectives [39, 40, 41, 42, 43, 44, 45, 46]. While the training objectives are very different, these methods enable strong zero-shot and linear-probe performance across a diverse set of tasks, suggesting that these methods effectively learn the *underlying statistics* of visual input. While these methods lack explicit dense supervision, the dense features from these models have been shown to have strong zero-shot emergent behavior with language-based segmentation [47], object part correspondence [48], and structural understanding [49].

**Artifacts in Vision Transformer Dense Features.** Recent work on DINOv2 [46] has found that Vision Transformers can have artifacts in their dense features. It has been proposed that artifacts can be mitigated with "register" tokens [50, 51, 52, 53]. These register tokens are effectively randomly initialized embeddings that are analogous to the [CLS] token. While registers participate in the self-attention process, they are discarded during output. This approach requires a model to be trained from scratch. The nature and the mechanisms that cause the emergence of artifact tokens are unclear, and there exists conflicting results on what information (global or no information) these artifact tokens contain [50, 54]. Recent work has further investigated the mechanism of register

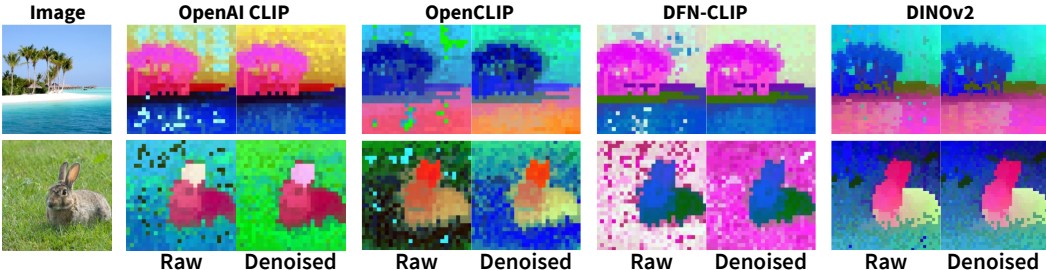

| Image | OpenAI CLIP | OpenCLIP | DFN-CLIP | DINOv2 |
|---|---|---|---|---|
| | Raw  Denoised | Raw  Denoised | Raw  Denoised | Raw  Denoised |

Figure 3: **Denoising Teacher Representations with Augmentations.** For each model, we visualize the UMAP of dense features before and after applying test-time augmentation. The results show that our proposed method produces noticeably cleaner dense feature representations without requiring gradient-based learning. Please zoom in for details.

tokens [55]. Our own results have found that artifact tokens are not necessarily high-norm, and can be low-norm as well. Unlike the observation by [33], we find that positional embeddings alone cannot account fully for the artifacts. Regardless of "why" artifact tokens emerge, removing these artifacts is an active area of research, with proposals based on registers [50], magnitude smoothness priors [54], and the foundational work on leveraging neural fields to denoise ViTs with a static artifact component [33, 56, 57]. A concurrent line of work has sought to remove artifacts for open-vocabulary segmentation with training-free attention modifications [47, 58, 59, 60, 61, 62]. Our framework can be applied to existing pretrained networks, introduces minimal additional parameters, can be applied to tasks beyond open-vocabulary segmentation, and makes no assumptions on the magnitude or static nature of the artifacts.

## 3 Methods

In this section, we will describe the PH-Reg framework, which we illustrate in Figure 2. This framework enables *existing pretrained ViTs* to benefit from register tokens, yielding significantly cleaner dense representations. During training, PH-Reg requires only unlabeled images for the self-distillation process. In section 3.1, we will first describe the denoising process of teacher network outputs. Unlike prior work that rely on a neural field/hash-grid, this method denoises dense features without the use of expensive gradient-based learning. In section 3.2, we will describe how we initialize and modify the student architecture. This approach only introduces a small set of additional parameters to the network. Finally in section 3.3 we will describe our distillation process.

### 3.1 Efficient Denoising of Teacher Representations

Our denoising process starts from the observation that artifact tokens are not static relative to image content. Put another way, if an image is shifted by a certain amount (with the gaps padded with whitespace), the artifacts do not shift by the same amount. As shown in Figure 2, given an RGB image $\mathcal{I} \in \mathbb{R}^{H \times W \times 3}$, we randomly sample $n$ random augmentation parameters $(\theta_1, \theta_2, ..., \theta_n)$, where each $\theta_i$ defines a horizontal/vertical offset $(x_i, y_i)$ and boolean $\text{flip}_i \in \{0, 1\}$ defined horizontally. For each image, we also compute the image space coordinate grid $\mathcal{C} = $ (x-coords, y-coords). Where $(x, y)$ are respectively in range $[0, 1]$: $x$ defines the left-right axis, and $y$ defines the top-down axis. The coordinates $\mathcal{C}$ help us keep track of the original location of an image region after augmentation. In practice, as we are working with a ViT model with patch size $k \times k$, where the tokenization process yields a $(\frac{H}{k}, \frac{W}{k})$ grid of image tokens, we define our parameters using offsets that are integer multiples of $k$ to facilitate efficient indexing. Together, the image $\mathcal{I}$, the coordinates $\mathcal{C}$, and augmentation parameter $\theta_i$ are provided to transform function to yield an augmented image $\mathcal{I}_i$ and new coordinates $\mathcal{C}_i$: $\mathcal{T}(\mathcal{I}, \mathcal{C}, \theta_i) \Rightarrow (\mathcal{I}_i, \mathcal{C}_i)$.

---

**Algorithm 1 Denoising Process**

**Input:** Image $\mathcal{I} \in \mathbb{R}^{H \times W \times 3}$;
Image space coordinates $\mathcal{C} \in [0, 1] \times [0, 1]$;
Augmentation parameters $\theta_1, \theta_2, ..., \theta_n$;
Augmentation function $\mathcal{T}$;
ViT teacher model $f_{teacher}$;
1. Zero init clean feature tensor $Q$
2. Zero init count tensor $K$
3. **For** i in {1, ..., n}:
4.     $\theta_i = (x_i, y_i, \text{flip}_i)$
5.     $(\mathcal{I}_i, \mathcal{C}_i) = \mathcal{T}(\mathcal{I}, \mathcal{C}, \theta_i)$
6.     Dense feature $F_i = f_{teacher}(\mathcal{I}_i)$
7.     $(F_i^{\text{valid}}, \mathcal{C}_i^{\text{valid}}) = \mathcal{T}^{-1}(F_i, \mathcal{C}_i, \theta_i)$
8.     $Q[\mathcal{C}_i^{\text{valid}}] = Q[\mathcal{C}_i^{\text{valid}}] + F_i^{\text{valid}}$
9.     $K[\mathcal{C}_i^{\text{valid}}] = K[\mathcal{C}_i^{\text{valid}}] + 1$
10. **return** $Q/K$

---

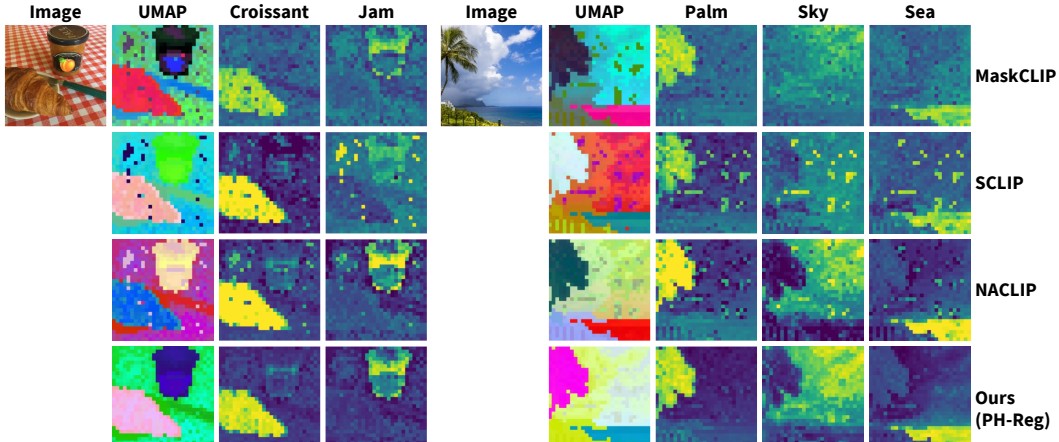

Figure 4: **Visualization of Open-vocabulary Semantic Segmentation.** We compare against MaskCLIP, SCLIP, NACLIP, and find that our method yields clean feature maps free of artifacts.

A teacher model $f_{\text{teacher}}$ is based on a frozen set of original network weights, without any additional parameters. Given an augmented image $\mathcal{I}_i$, this network outputs feature $F_i$. We restore the features to the original location within an image using the inverse transform function $\mathcal{T}^{-1}(F_i, \mathcal{C}_i)$. The restored features are additively accumulated across different augmentation parameters, while keeping track of the number of occurrences for each location. At the end, the dimension-wise sample mean is computed for the accumulated features. We present our full denoising process in Algorithm 1. The patch-wise expected value of this representation is the same as the optimal value when optimizing a discrete grid of representations to minimize mean squared error (as used in DVT [33] and traditional neural field based methods). However, as we do not require gradients, this denoising process can be done in less than *200ms*, roughly two magnitudes faster than neural field based denoising in DVT. The comparison between raw and denoised dense feature visualizations is shown in Figure 3.

### 3.2 Design of the Student Network

Our objective is to preserve maximimal computational efficiency of the student model, while leveraging the knowledge of the pre-trained weights. For this purpose we introduce $m$ number of register tokens, providing a minimally invasive enhancement to the base architecture. After the addition of register tokens, a total of $m + 1 + \frac{H}{k} \times \frac{W}{k}$ tokens participate in the self-attention process. Unlike prior work that trains registers from scratch [50], this approach updates only these registers and selectively unfreezes specific components during distillation, preserving the majority of the ViT's pretrained weights. Through ablation studies, we identify optimal unfreezing strategies, such as adjusting convolution layers, positional embeddings, or the last transformer block.

### 3.3 Learning and Optimization of the Student

We employ a multi-objective distillation strategy, combining cosine similarity and mean squared error losses to ensure both directional and magnitude alignment between teacher representations $F_{teacher}$ and student representations $F_{student}$. Our final loss is: $\text{Loss}_{\text{total}} = 1 - \texttt{cossim}(\text{target}, \text{predicted}) + \texttt{MSE}(\text{target}, \text{predicted})$.

## 4 Experiments

In this section, we comprehensively evaluate the performance of PH-Reg on a diverse set of dense tasks, first using a zero-shot setup for open-vocabulary segmentation in section 4.1, followed by linear probe based segmentation and depth tasks in section 4.2. Finally we perform ablation studies to explore design decisions and investigate the nature of artifacts across different models in section 4.3. All implementation details are provided in the appendix.

### 4.1 Open-vocabulary Semantic Segmentation Evaluation

**Datasets.** In this section, we follow prior works [59, 62, 61] to evaluate our approach on six semantic segmentation datasets, with their names abbreviated (in parentheses) to conserve table space: PASCAL

Table 1: **Open-vocabulary Semantic Segmentation Quantitative Evaluation Results on 8 Benchmarks.** While the first 3 benchmarks (VOC21, PC60 and Obejct) include a background class, the remaining benchmarks do not. We report the mean Intersection over Union (mIoU, %) metric, where higher values indicate better performance, for our method and all baseline models. The best result for each dataset is highlighted in **bolded**. Additional results are provided in the supplementary material.

| Method | VOC21 | PC60 | Object | VOC20 | PC59 | Stuff | City | ADE | Avg. |
|---|---|---|---|---|---|---|---|---|---|
| CLIP [34] | 18.60 | 7.84 | 6.50 | 49.05 | 11.17 | 7.19 | 6.65 | 3.16 | 13.77 |
| MaskCLIP [47] | 49.27 | 25.46 | 26.94 | 66.56 | 28.62 | 18.80 | 28.33 | 13.70 | 32.21 |
| SCLIP [59] | 59.62 | 31.74 | 33.52 | 81.53 | 34.46 | 22.65 | 32.34 | 16.45 | 40.08 |
| ClearCLIP [61] | 59.76 | 32.56 | 32.77 | **84.56** | 35.91 | 23.89 | 30.04 | 16.65 | 39.52 |
| NACLIP [62] | 58.88 | 32.20 | 33.15 | 79.70 | 35.16 | 23.30 | 35.48 | 17.42 | 39.41 |
| MaskCLIP + DVT [47, 33] | 44.29 | 25.08 | 20.89 | 65.88 | 29.50 | 17.10 | 30.89 | 14.06 | 30.96 |
| NACLIP + DVT [62, 33] | 60.25 | 32.73 | 32.89 | 80.26 | 35.91 | 23.41 | 36.31 | 17.54 | 39.91 |
| Ours (PH-Reg) | **63.01** | **34.52** | **35.27** | 83.05 | **37.88** | **24.66** | **37.17** | **19.22** | **41.85** |

VOC 2012 (VOC21) [63], PASCAL Context (PC 60) [64], COCO-Object (Object) [65], COCO-Stuff (Stuff) [66], Cityscape (City) [67], ADE20K-150 (ADE) [68]. In addition to these standard benchmarks, we also evaluate on two commonly used variants, PASCAL VOC 2012 (VOC20) and PASCAL Context (PC 59), in which the background class is excluded from the evaluation. For all experiments, we utilize the same evaluation harness for all methods, and apply a sliding window inference strategy for non-square images. We also resize input images such that the shorter side is fixed to specific resolutions, accommodating the varying original image sizes across datasets.

**Baselines.** We compare our method against several relevant approaches in open-vocabulary semantic segmentation, including MaskCLIP [47], SCLIP [59], ClearCLIP [61], and NACLIP [62]. We also include vanilla CLIP as a baseline in our comparison, as it can be adapted for semantic segmentation. Unless otherwise specified, all visual encoders use the widely adopted pretrained ViT backbone with the same `OpenAI CLIP ViT-B/16` weights to ensure a fair comparison. We also include denoised versions of MaskCLIP and NACLIP produced by DVT, as DVT represents a closely related method to our approach. Unless otherwise noted, for CLIP we adopt the most basic MaskCLIP framework (direct $v$ output without any attention modifications) as our student model. Notably, we re-implemented all baselines using the same prompt templates as in [59, 62]. All reported results are obtained without any post-processing refinement.

**Quantitative Results.** Table 1 summarizes the quantitative comparison results of various open-vocabulary semantic segmentation models. We observe that PH-Reg CLIP consistently outperforms all compared methods on 7 out of 8 evaluated benchmarks, with particularly strong results in VOC21 (63.01%) and COCO Object (35.27%). Moreover, PH-Reg CLIP surpasses the denoised versions of MaskCLIP and NACLIP, where DVT fails to yield significant performance gains. We believe this is caused by the residual estimator in DVT, which assumes stationary artifacts – an assumption that does not hold consistently for training-free open vocabulary segmentation methods based CLIP. We note that ClearCLIP slightly outperforms our method on the VOC20 dataset. This may be attributed to its use of correlative self-attention, *i.e.* $q$-$q$ attention, which incorporates feature localization cues. This explanation is plausible, as SCLIP, which also employs $q$-$q$ attention, similarly outperforms NACLIP on the VOC20 dataset.

**Qualitative Results.** Figure 4 presents a qualitative comparison between our PH-Reg CLIP and three baseline models: MaskCLIP, SCLIP, and NACLIP. We visualize the UMAP of the dense features produced by each model, as well as the corresponding heatmaps generated from different text queries. Our qualitative observations are as follows: **1.** Artifact tokens are frequently observed in the UMAP visualizations of MaskCLIP, SCLIP, and NACLIP. While some artifact tokens are reduced in the heatmap of MaskCLIP, they remain prevalent in the heatmaps of both SCLIP and NACLIP. **2.** The presence of artifact tokens hinders the models' ability to maintain fine-grained spatial alignment with the text queries, leading to suboptimal localization. **3.** In contrast, PH-Reg CLIP consistently produces cleaner UMAPs and more fine-grained, semantically aligned heatmaps, which correspond well to meaningful local image structures. These visualizations demonstrate that our method effectively preserves the consistency of dense feature representations and enhances semantic alignment between visual and textual modalities.

Table 2: **Linear Probe Based Evaluation Results on Segmentation and Depth.** PH-Reg improves pretrained ViT backbones across various dense prediction tasks. For semantic segmentation, we report the mean Intersection over Union (mIoU, %) metric and mean accuracy (mAcc, %). For monocular depth estimation, we report Root Mean Squared Error (RMSE), absolute relative error (Abs Rel), and accuracy under threshold $\delta_1$. The best result for each dataset is highlighted in **bold**.

| Method | VOC21 | | ADE | | NYUd | | |
|---|---|---|---|---|---|---|---|
| | mIoU($\uparrow$) | mAcc($\uparrow$) | mIoU($\uparrow$) | mAcc($\uparrow$) | RMSE($\downarrow$) | Abs Rel($\downarrow$) | $\delta_1$($\uparrow$) |
| CLIP [34] | 73.88 | 83.37 | 35.78 | 47.3 | 0.6843 | 0.2115 | 64.93 |
| CLIP + DVT [34, 33] | 74.74 | 84.33 | 36.39 | 48.14 | 0.6800 | 0.2089 | 65.07 |
| NACLIP [62] | 74.01 | 83.16 | 37.06 | 48.33 | 0.6852 | 0.2082 | 64.52 |
| NACLIP + DVT [62, 33] | 74.47 | 82.98 | 36.91 | 48.56 | 0.6845 | 0.2122 | 65.11 |
| MaskCLIP [47] | 70.28 | 79.06 | 34.43 | 44.74 | **0.6645** | 0.2030 | 67.71 |
| MaskCLIP + DVT [47, 33] | 71.38 | 80.49 | 34.43 | 44.86 | 0.6792 | 0.2091 | 64.96 |
| Ours (PH-Reg) | **75.32** | **84.96** | **38.07** | **49.58** | 0.6746 | **0.1995** | **68.17** |
| OpenCLIP [70] | 71.31 | 80.64 | 37.68 | 49.8 | 0.6853 | 0.2113 | 64.86 |
| OpenCLIP + DVT [70, 33] | 72.58 | 83.42 | 38.30 | 50.91 | 0.6811 | 0.2159 | 64.73 |
| OpenCLIP + Ours | **73.25** | **83.99** | **39.32** | **51.24** | **0.6784** | **0.2019** | **65.32** |
| DFN-CLIP [71] | 71.98 | 82.07 | 36.81 | 47.83 | 0.6860 | 0.2118 | 64.50 |
| DFN-CLIP + DVT [71, 33] | **73.09** | **83.52** | 37.73 | 49.39 | 0.6852 | 0.2092 | 64.65 |
| DFN-CLIP + Ours | 72.97 | 82.48 | **39.15** | **50.61** | **0.6768** | **0.2052** | **65.26** |
| DINOv2 [46] | 84.13 | 92.00 | 47.82 | 60.50 | 0.4566 | 0.1391 | 82.92 |
| DINOv2 + DVT [46, 33] | **85.43** | **93.37** | **48.86** | **61.61** | 0.4329 | 0.1289 | 85.23 |
| DINOv2 + Ours | 84.85 | 92.46 | 48.66 | 61.57 | **0.4306** | **0.1216** | **86.35** |

## 4.2 Linear Probe Based Segmentation & Depth Evaluation

**Datasets & Baselines.** In this section, we evaluate our approach in two semantic segmentation datasets: PASCAL VOC 2012 (VOC21) [63] and ADE20K-150 (ADE) [68] and one monocular depth estimation dataset: NYUv2-Depth dataset (NYUd) [69]. Since DVT [33] targets a similar objective with our approach, we include both the vanilla models and their denoised versions produced by DVT as comparison baselines. We adopt the same linear probe experimental setup as in [33], and train a linear layer integrated into the backbones, as a decode head to predict pixel-wise segmentation or depth logits from patch tokens. Table 2 summarizes the main experiment results.

**Semantic Segmentation Results.** As shown in Table 2, We observe significant and consistent improvements, outperforming at least 4 out of 6 denoised ViT backbones across the evaluated datasets. While DVT consistently enhances the performance of DINOv2 and vanilla CLIP, it provides only limited improvements for other ViT backbones derived from other CLIP models. In contrast, our approach yields substantial performance boosts across these backbones, especially a notable +5.04% mIoU on VOC21 and +3.64% mIoU on ADE20k. These results demonstrate that our method can be robustly adopted to enhance the performance of diverse ViT backbones in semantic segmentation.

Notably, DVT relies on neural fields and requires gradient-based optimization, making the iterative denoising process applied to each image individually highly time-consuming. Our method leverages test-time augmentation for denoising, enabling the generation of cleaner dense feature representations without incurring excessive computational overhead. However, our results also show that the residual estimator as introduced in DVT may be beneficial to some model types (DINOv2, DFN-CLIP) more so than others. These results highlight that PH-Reg achieves superior performance in suppressing artifact tokens through a more robust and efficient design.

**Depth Estimation Results.** Following prior work [46, 33] we adopt AdaBins [72] for monocular depth evaluation. As shown in Table 2, our method consistently improves the performance of pretrained ViT backbones whereas the DVT assumption of stationary artifacts mostly hold true for DINOv2. Additionally, DVT achieves performance gains using an additional transformer block with 0.08× the parameters of the base models [33], our method achieves superior results with only a negligible increase in parameter count introduced by the register tokens. These results demonstrate the efficiency of our approach, yielding noticeable performance gains with minimal model overhead.

## 4.3 Ablation Studies and Investigation of Artifacts

In this section, we conduct ablation studies on `OpenAI CLIP ViT-B/16` to investigate various architectural and training components, focusing on both model performance and training feasibility.

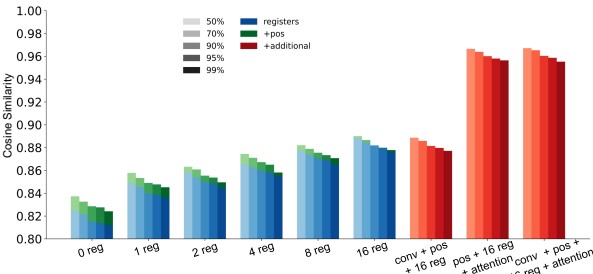 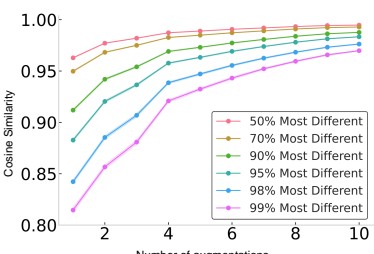

(a) **Registers' behavior.** This plot illustrates adding registers improve PH-Reg teacher performance. In the blue settings, only registers are unfrozen. The green settings represent the improvements when positional embeddings are unlocked additionally. The red settings represent performance of unlocking more layers.

(b) **Augmentations improves cosine similarity.** The plot illustrates how increasing the number of augmentations improves the alignment of the model's predictions with the target of 200 augmentations' features, as measured by cosine similarity.

Figure 5: **Ablation on number of registers and augmentations**

**The Number of Register Tokens.** We evaluate the influence of the number of register tokens on the cosine similarity between the student model's outputs and the target values using the COCO Caption dataset [73]. Specifically, we distill the student model with 0, 1, 2, 4, 8, or 16 register tokens.

As illustrated in Figure 5a, the cosine similarity increases as the number of register tokens grows, indicating improved alignment with the target representations. However, the performance gain becomes marginal when increasing the number of registers from 4 to 8 and from 8 to 16. Based on this observation, we use 16 register tokens in all subsequent experiments.

**Distillation Architectural Settings.** We further evaluate the impact of architectural configurations during distillation by analyzing cosine similarity on the COCO Caption dataset. In this setting, we vary the number of register tokens from 0 to 16 while allowing the positional embeddings to be updated during training. As shown in Figure 5a, the improvement in the cosine similarity from unlocking the position embedding becomes less pronounced as the number of register tokens increases. Nonetheless, unlocking positional embeddings continues to provide a positive effect on alignment. This result suggests that in contrast to DVT, the positional embedding itself is unlikely to fully explain the artifact tokens.

Next, we fix the number of register tokens to 16 and evaluate the effect of unlocking additional layers, including the convolutional patch embedding layer and the later attention layers. For all experiments, we report *50th*, *70th*, *90th*, *95th*, and *99th* percentiles (of cosine similarity to capture the distribution of the most dissimilar features.) Our analysis reveals that incorporating even a single register leads to substantial improvements. In particular, the *99th* percentile of feature cosine similarity in the 1-register configuration exceeds the *50th* percentile (median) of the raw case without registers. This indicates that registers significantly enhance the quality of feature representations across the distribution, not only in extreme cases.

As suggested by [54, 33], attention layers close to the output also play an important role, which we confirm in our experiments. As shown in Figure 5a, unlocking the last attention layer significantly increases the cosine similarity between the student model's outputs and the target values. While unlocking the convolutional patch embedding layer alone slightly reduces cosine similarity value, the overall value improves when both the convolutional patch embedding and later attention layers are unlocked, compared to the baseline with only unlocked the position embeddings and later attention layers with 16 registers. Therefore, we unlock the positional embeddings, the convolutional patch embedding layer, and the final attention layer during distillation.

**The Number of Augmentations in the Denoising Process.** We evaluate our approach using cosine similarity on the COCO Caption dataset to investigate the effect of the number of augmentations. The

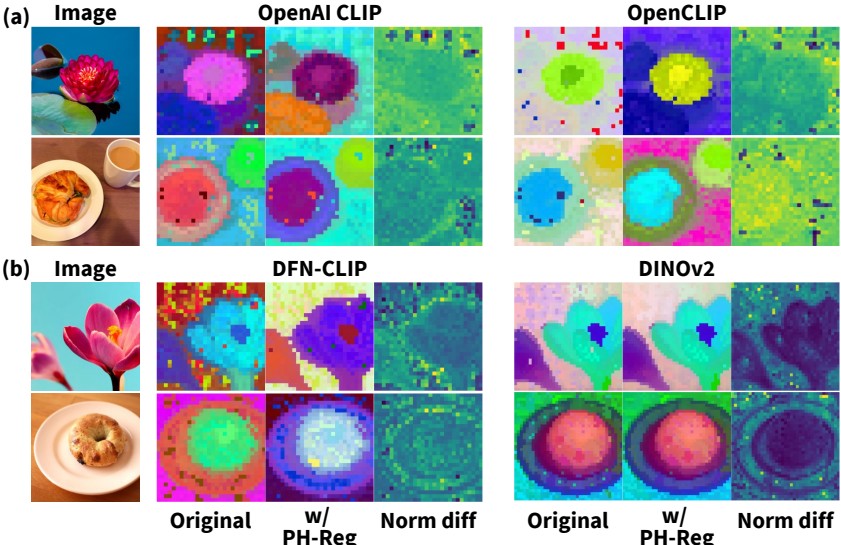

Figure 6: **Comparison of Original and PH-Reg Features and Norms.** While prior work has noted artifact tokens in DINOv2 as having higher norm than other tokens, we observe this is not the case for all models. Some models have artifact tokens with lower magnitude.

Table 3: **Distillation Approaches.** This table reports the contributions of different components in our distillation framework. *Vanilla* refers to the student MaskCLIP, *Denoising Only* refers to MaskCLIP with 10x averaging, the other rows refer to using NACLIP as teacher and MaskCLIP as student.

| Approach | VOC21 | PC60 | Object | VOC20 | PC59 | Stuff | City | ADE | Avg. |
|---|---|---|---|---|---|---|---|---|---|
| Vanilla | 49.27 | 25.46 | 26.94 | 66.56 | 28.62 | 18.80 | 28.33 | 13.70 | 32.21 |
| Denoising only (10x aug) | 51.41 | 28.13 | 29.00 | 69.58 | 31.03 | 20.25 | 31.82 | 15.20 | 34.55 |
| Distill, no reg, no denoise | 61.16 | 33.51 | 34.51 | 81.51 | 36.70 | 23.96 | 35.74 | 18.34 | 40.68 |
| Distill, with reg, no denoise | 61.27 | 33.52 | 34.39 | 81.52 | 36.74 | 23.92 | 35.55 | 18.38 | 40.66 |
| Distill, no reg, with denoise | 62.48 | 34.28 | 35.00 | 82.27 | 37.62 | 24.46 | 36.83 | 18.92 | 41.48 |
| Full Pipeline | **63.01** | **34.52** | **35.27** | **83.05** | **37.88** | **24.66** | **37.17** | **19.22** | **41.85** |

student model is distilled with 1 to 10 augmentations, where one of them is always an unmodified image. As shown in Figure 5b, a high convergence threshold is observed at the *99th* percentile where even the most dissimilar cases exhibit cosine similarity values above 0.95, indicating a substantial reduction in feature space outliers. As a result, we employ 10 augmentations to generate high-quality features efficiently, thereby reducing computational overhead.

**Distillation Approaches.** We evaluate the contribution of different components in our distillation approach by conducting open-vocabulary semantic segmentation task on 8 benchmarks. In this setting, the student model is distilled by sequentially removing each component from our approach. These components include the denoising process with 10 augmentations, 16 additional register tokens, and self-distillation. As shown in Table 3, each component contributes to improving the student model's performance. By comparing our full pipeline, where student model is distilled with denoising process and registers, we find that approximately half of the improvement comes from registers, and the other half results from denoising process applied to the teacher model.

**The Ratio of Shifting in the Denoising Process.** To investigate the impact of different shifting ratios in our denoising process, we conduct open-vocabulary semantic segmentation task on 8 benchmarks. We gradually increase the shifting ratio from 0% to 30%, while fixing the number of augmentation at 10. As shown in Table 4, a shifting ratio of 10% demonstrates the best performance across 4 datasets, while a shifting ratio of 15% achieves the best performance across 3 datasets and provides optimal average performance across all 8 datasets. Therefore, we adopt a shifting raio of 15% in denoising process when applied to the teacher model.

Table 4: **Effect of Shifting Ratio.** This table reports the impact of different shifting ratios used in the denoising process, as defined in 3.1. Throughout this experiment, we use `OpenAI CLIP ViT-B/16` as the backbone for the model and apply 10 augmentations in the denosing process.

| Ratio | VOC21 | PC60 | Object | VOC20 | PC59 | Stuff | City | ADE | Avg. |
|---|---|---|---|---|---|---|---|---|---|
| 0% | 58.88 | 32.20 | 33.15 | 79.70 | 35.16 | 23.30 | 35.48 | 17.42 | 39.41 |
| 10% | **60.49** | **32.91** | **33.73** | **80.47** | **35.94** | 23.86 | 36.60 | 17.94 | 40.24 |
| 15% | 60.47 | 32.89 | 33.61 | 80.36 | **35.94** | **23.89** | 36.87 | **18.10** | **40.27** |
| 20% | 60.39 | 32.82 | 33.46 | 79.93 | 35.85 | 23.86 | 36.98 | 18.07 | 40.17 |
| 25% | 60.26 | 32.75 | 33.26 | 79.75 | 35.75 | 23.78 | 37.02 | 18.00 | 40.07 |
| 30% | 60.14 | 32.76 | 33.25 | 79.39 | 35.77 | 23.77 | **37.07** | **18.10** | 40.03 |

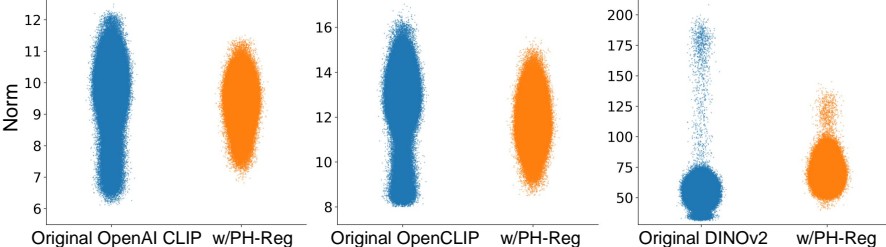

Figure 7: **Patch Norms.** This figure illustrate norms of patch tokens of different backbones. Our method effectively reduces the variance of token norms and reduces the outliers, regardless if the artifacts are lower/higher norm.

### 4.4 Investigation of Artifacts and Registers

While prior work has noted that artifact tokens are high magnitude in DINOv2 [50], in Figure 6 we find that this is not always the case. In OpenAI CLIP and OpenCLIP, the artifacts are generally lower norm than their surrounding patches. In contrast, in DFN-CLIP and DINOv2, the artifacts are higher norm. This illustrates that there may be elements of the training dynamic at play, as the artifact norms can differ even when the training objective is very similar. In Figure 7 we visualize the norms of the original network and those with registers added, we find that our method effectively reduces the variance of the patch norms.

## 5  Discussion

**Limitations and Future Work**    In this work we proposed PH-Reg, a method to reduce the artifact tokens in existing pre-trained vision transformers. We show that PH-Reg can eliminate artifact tokens in ViTs effectively and generate clean dense feature maps, enhancing the performance in downstream dense prediction tasks. This approach relies on test time augmentation to denoise dense feature presentations in the teacher model. While our method generally outperforms DVT on CLIP based models, we sometimes underperform when using the DINOv2 backbone. We believe this is due to the static artifact estimator present in DVT. The assumption of static artifacts holds true for some models (DINOv2), but not for others (CLIP). A potential avenue for additional investigation is how to dynamically determine the artifacts without strong stationary assumptions.

**Conclusion**    We introduce a novel post-training method PH-Reg, for learning clean dense feature representations in ViTs through an efficient self-distillation framework that does not require additional labeled data. Our approach leverages test-time augmentation to denoise the teacher model, and guide the student model to optimize the dense feature representations. This enables us to eliminate artifact tokens effectively by integrating learnable registers into existing pretrained models, without the need for training from scratch. We demonstrate that the distilled ViTs generate fine-grained dense feature maps, enhancing the consistency of feature representations in ViTs. We further show that cleaner dense feature maps in ViTs leads to quantifiable improvements on dense prediction tasks. Finally, we illustrate that the distilled ViTs can accurately capture meaningful semantic structures in images, as shown by heatmaps generated from CLIP text queries. We validate our conclusions with extensive evaluations across multiple dense prediction benchmarks.

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

## A    Technical Appendices and Supplementary Material

# Sections

1. Implementation Details for Self-Distillation (section B)
2. Implementation Details for Quantitative Evaluation (section C)
3. Additional Efficiency Analysis of PH-Reg Compared to DVT (section D)
4. Additional Qualitative Examples for Segmentation (section E)
5. Additional Qualitative Heatmaps for PH-Reg Zero-Shot (section F)
6. Proof of Test Time Augmentation (section G)

# B    Implementation Details for Self-Distillation

In this section, we provide a detailed overview of how we implement self-distillation in PH-Reg. Our self-distillation framework consists of one teacher model and one student model. While both the teacher and student model are initialized from the same weights, the teacher is frozen, while additional register parameters are added to the student network.

Our codes and weights are avaible in https://github.com/0raiser0/PH-Reg.git.

## B.1    Model Architectures

**Teacher Model Architecture.** For CLIP based models, since we focus on zero-shot open-vocabulary segmentation, we utilize the NACLIP modification to the final layer. This modification does not introduce any additional weights to the teacher network, and is training-free. Our empirical analysis in Section B.4 shows that NACLIP's neighborhood attention mechanism improves feature consistency. For DINOv2, we directly use the final output layer, without any modification to the teacher network.

**Student Model Architecture.** Based on the results from the ablation studies we integrate 16 register tokens into the student model. For the CLIP based student, to ensure representational alignment we directly take the $v$ head from the output layer (the MaskCLIP output). For DINO based students, we do not apply such modifications. We bicubicly upsample the positional embedding so it matches the input image. Unless otherwise specified, this modification is applied consistently, while all other layers remain unchanged.

## B.2    Model Implementation

Table S.1: **Model Implementation Libraries and Weights.** We compare models trained using different datasets and objectives.

| Model | Library | Weight |
|---|---|---|
| CLIP | clip (OpenAI) | ViT-B-16 |
| OpenCLIP | open_clip | hf-hub:laion/CLIP-ViT-B-16-laion2B-s34B-b88K |
| DFN-CLIP | open_clip | hf-hub:apple/DFN2B-CLIP-ViT-B-16 |
| DINOv2 | transformers (Hugging Face) | facebook/dinov2-base |

We provide the model weights and the corresponding implementation libraries in Table S.1.

## B.3    Optimization

In the distillation process, the shorter side of each input image is resized using bicubic interpolation to 448 for CLIP-based models and 518 for DINOv2. The resized image is then randomly cropped into a square of size (448, 448) or (518, 518), respectively. For each input image, we generate $N = 10$ augmentations using random shifts and horizontal flips. Assuming an image length of 1, we uniformly sample the shift for both the horizontal and vertical axes from $[-0.15, 0.15]$. While the horizontal flip is sampled with probability 0.5. To ensure each patch is covered, we do not apply any augmentation to the first image of the 10. All shifted images are concatenated and fed into the teacher model, while the original (unshifted) images are used as input to the student model. The target feature is computed as the average of these 10 augmentations. To accommodate the resized input images for both the teacher and student models, we consistently resize the positional embeddings using bicubic interpolation. During training, the weights of the teacher model are frozen. In the student model, we allow updates to registers, the positional embeddings, the convolutional patch embedding layer, and the final transformer layer containing the self-attention mechanism.

The distillation framework is implemented in PyTorch, with distributed training managed via PyTorch Accelerate. Training is conducted on 4 NVIDIA Ada 6000 GPUs, with mixed-precision optimization to balance computational efficiency and numerical stability. Detailed training configurations are provided in Table S.2 and Table S.3.

Table S.2: Configs for CLIP-based models.

| Config | Value |
|---|---|
| optimizer | AdamW |
| initial learning rate | 3e-4 |
| final learning rate | 1e-5 |
| weight decay | 1e-2 |
| optimizer momentum $(\beta_1, \beta_2)$ | (0.9, 0.999) |
| learning rate scheduler | Exponential Scheduler |
| batch size | 16 |
| training epochs | 100 |
| augmentation | RandomSquareCrop |

Table S.3: Configs for DINOv2.

| Config | Value |
|---|---|
| optimizer | AdamW |
| initial learning rate | 1e-4 |
| final learning rate | 5e-6 |
| weight decay | 1e-2 |
| optimizer momentum $(\beta_1, \beta_2)$ | (0.9, 0.999) |
| learning rate scheduler | Exponential Scheduler |
| batch size | 8 |
| training epochs | 100 |
| augmentation | RandomSquareCrop |

## B.4 Pearson Analysis of Open-vocabulary Segmentation

Table S.4: **Open-vocabulary Semantic Segmentation Quantitative Comparison on 7 datasets.** We report the Pearson correlation coefficient for the zero-shot query against the one-hot ground truth labels. The results are averaged within each image, then averaged across images. Compared to mIoU, pearson does not require knowledge of all of the categories present an image (via softmax). The value ranges from -1 to 1, where 1 = perfect positive correlation, -1 = perfect negative correlation, and 0 = no linear correlation. The best result for each dataset is highlighted in **bolded**.

| Method | VOC21 | PC60 | VOC20 | PC59 | Stuff | City | ADE20k | Avg. |
|---|---|---|---|---|---|---|---|---|
| SCLIP | -0.005 | 0.349 | 0.409 | 0.443 | 0.323 | 0.291 | 0.308 | 0.303 |
| ClearCLIP | 0.012 | 0.428 | 0.489 | 0.543 | 0.393 | 0.336 | 0.418 | 0.374 |
| NACLIP | 0.011 | 0.422 | 0.470 | 0.543 | 0.392 | 0.363 | 0.425 | 0.375 |
| NACLIP+DVT | 0.003 | 0.438 | 0.487 | 0.551 | 0.395 | 0.367 | 0.427 | 0.381 |
| Ours (PH-Reg) | **0.013** | **0.468** | **0.494** | **0.590** | **0.424** | **0.381** | **0.461** | **0.404** |

In this section we present additional evaluation results on open-vocabulary semantic segmentation via the pearson metric. Results are illustrated in Table S.4. Overall, PH-Reg CLIP significantly outperforms the baseline models on 7 datasets. Even in the absence of prior category knowledge, PH-Reg CLIP achieves an average performance of 0.404, representing a clear improvement over the second-best method, DVT enhanced NACLIP, with an average performance of 0.381. These results highlight that our approach improves the consistency of dense feature representations by reducing artifact tokens, thereby offering a robust and generalizable enhancement over existing methods.

We further observe that both ClearCLIP and NACLIP achieve competitive results; however, NACLIP significantly outperforms ClearCLIP on ADE20K and Cityscapes. The former requires the model to handle a large number of categories, while the latter demands fine-grained localization of small objects. Based on this observation, we choose NACLIP as our primary teacher model, leveraging its neighbor attention mechanism to enhance the student model's performance on these challenging tasks.

Table S.5: **Dataset Specific Details for Open-vocabulary Semantic Segmentation.** We list the per-dataset resolution, crop size, and stride used for each dataset. We maintain the same settings for all methods within a given dataset.

| Dataset | VOC21 | PC 60 | Object | VOC20 | PC 59 | Stuff | City | ADE |
|---|---|---|---|---|---|---|---|---|
| Resize resolution | 448 | 448 | 336 | 336 | 448 | 448 | 560 | 448 |
| Crop size | 336 | 336 | 336 | 336 | 336 | 336 | 224 | 336 |
| Stride | 112 | 112 | 112 | 112 | 112 | 112 | 112 | 112 |

Table S.6: **Training Time Efficiency Analysis.** We report the training time of our PH-Reg method compared to DVT. For fairness, we restrict all stages of our model to a single GPU and evaluate the same model (DINOv2 ViT-B).

| Method | Stage 1 Extraction | Stage 1 Distillation | Stage 2 Training | Total |
|---|---|---|---|---|
| DVT | 2998 min | 18340 min | 570 min | 21908 min (365.1 h) |
| PH-Reg | - | - | - | 9000 min (150 h) |

# C   Implementation Details for Quantitative Evaluation

In this section, we provide detailed implementation information for our quantitative evaluation experiments. In section C.1, we present the evaluation details for open-vocabulary semantic segmentation (OVSS). In section C.2, we describe the evaluation details for linear probe based semantic segmentation and monocular depth estimation.

## C.1   Implementation Details of Open-vocabulary Semantic Segmentation.

We follow SCLIP and NACLIP in the setup for the open-vocabulary semantic segmentation evaluation. For fairness, we utilize the same parameters for all models. We resize input images such that the shorter side is scaled to a specific resolution, while maintaining the original aspect ratio for the longer side. Additionally, we set fixed crop sizes and strides during evaluation. All evaluation parameters are summarized in Table S.5, while all other settings follow their default configurations.

## C.2   Implementation Details of Linear Probe Based Evalution.

Our linear probe evaluation follows prior works (Vision Transformers Need Registers, Denoising Vision Transformers), where a linear layer is trained as a decoding head to predict pixel-wise segmentation or depth logits.

**Semantic Segmentation.** We extract the final output features from the frozen backbone and, if applicable, pass them through the denoiser (for the DVT baseline). A single learnable linear layer is then trained to predict the segmentation logits. For CLIP-based models, both training and testing images are resized to (448, 448), while for DINOv2, the images are resized to (518, 518).

**Monocular Depth Estimation.** Similar to semantic segmentation, we extract features from the backbone, and pass them through the denoiser if applicable. Following the method in DVT and DINOv2, we then append the **[CLS]** token to each patch token to enrich the feature representations for all methods. A linear layer is trained using SigLoss and gradient loss (scaled by a factor of 0.5) to predict depth values into 256 uniformly distributed bins. We adopt DVT's learning rate of 5e-3 for all experiments.

# D   Additional Efficiency Analysis of PH-Reg Compared to DVT

In this section, we provide a detailed efficiency analysis of PH-Reg compared to DVT, focusing on three aspects: training time, space usage, and inference cost.

**Training Time.** When evaluating training time, we utilize the official code provided in the DVT repository without any modification. The original DVT code specifies a single GPU for feature

Table S.7: **Inference Efficiency Analysis.** We report the inference cost results for all models, evaluating efficiency based on model size and GFLOPs.

| Method | GFLOPs | Params (M) |
|---|---|---|
| MaskCLIP | 62.89 | 86.19 |
| NACLIP | 64.76 | 86.19 |
| NACLIP w/ denoising (10x) | 647.6 | 86.19 |
| NACLIP + DVT | 70.32 | 94.07 |
| CLIP + PH-Reg (Ours) | 64.16 | 86.66 |

extraction and learning – for fairness we also limit all stages of our own model to a single GPU here. For DVT and PH-Reg, we evaluate the same model (DINOv2 ViT-B). Results are illustrated in Table S.6. Our method has the advantage of not utilizing gradient-based neural field learning as done in DVT. Therefore, our method trains the student model in a single stage, saving over 58.9% of the time compared to DVT.

**Space Usage.** A further advantage is the space utilization during training. DVT requires saving 1.4 terabytes of intermediate neural fields in the form of serialized Instant-NGP files. Our method computes all distillation targets on the fly, and requires no additional space.

**Inference Cost.** When evaluating testing cost for all models, we cast parameters to *fp32* dtype, and use eager attention implementation for all models. For our method, we include the positional embeddings adapted for 448 resolution. For MaskCLIP and NACLIP, current official implementations have the same number of parameters as the original CLIP, although a small reduction can be achieve in MaskCLIP by discarding the last *q, k* head. For DVT, we evaluate their stage 2 model, corresponding to the transformer block denoiser coupled to a original vision transformer. Results are illustrated in Table S.7. Our method utilizes approximately 10% fewer FLOPs and 10% fewer parameters during inference compared to DVT.

# E  Additional Qualitative Examples

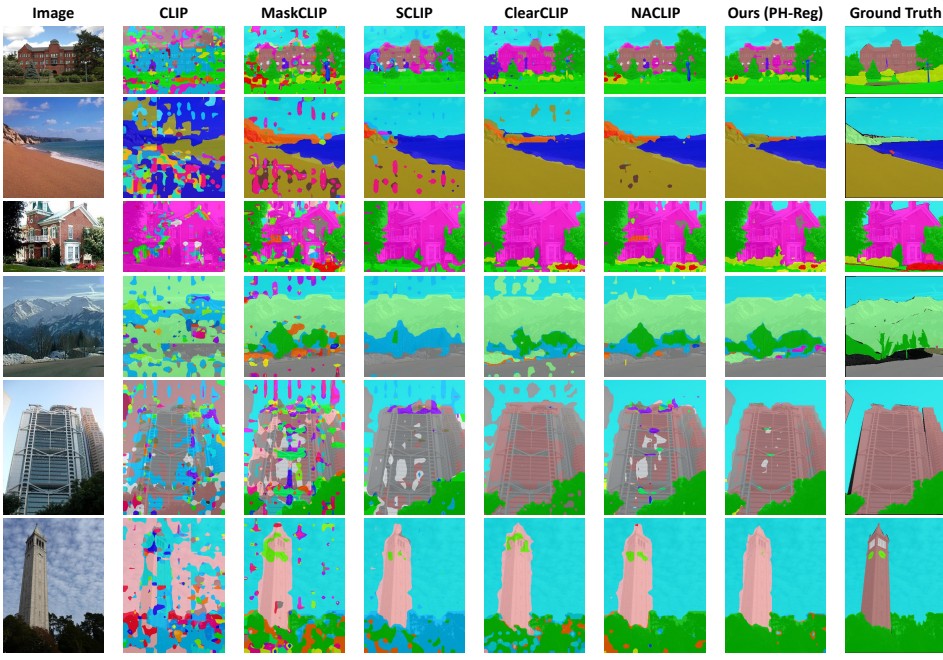

Figure S.1: Open-vocabulary semantic segmantation qualitative comparision between different baseline models on ADE20K.

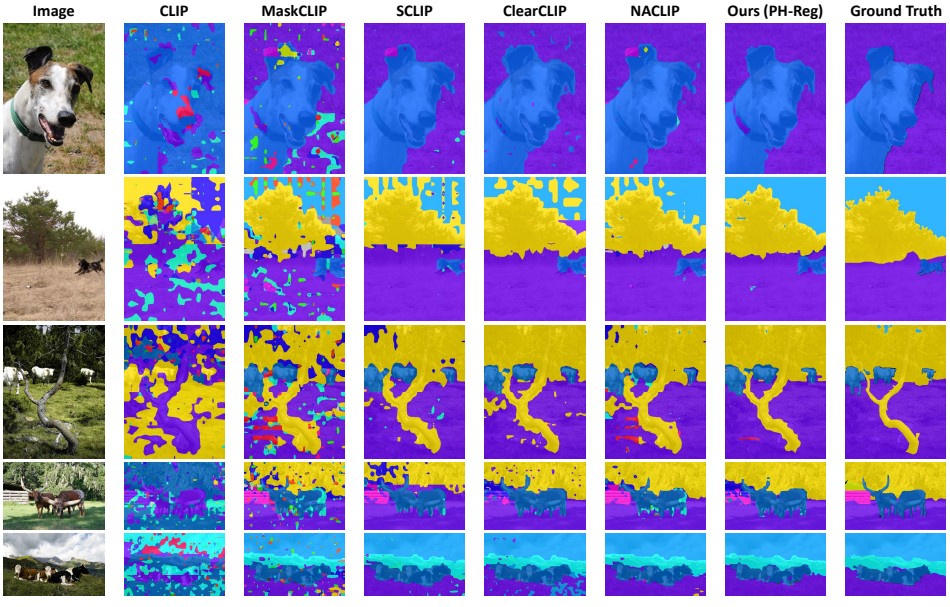

Figure S.2: Open-vocabulary semantic segmantation qualitative comparision between different baseline models on Pascal Context59.

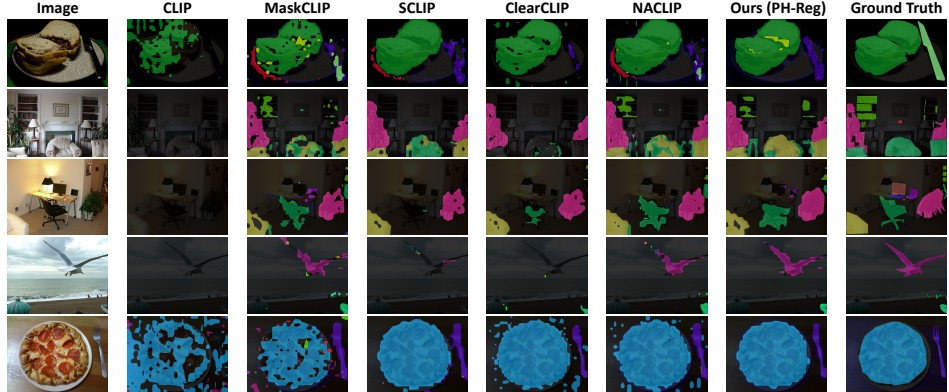

Figure S.3: Open-vocabulary semantic segmantation qualitative comparision between different baseline models on COCO Obejct.

# F   Additional Qualitative Heatmaps for PH-Reg Zero-Shot

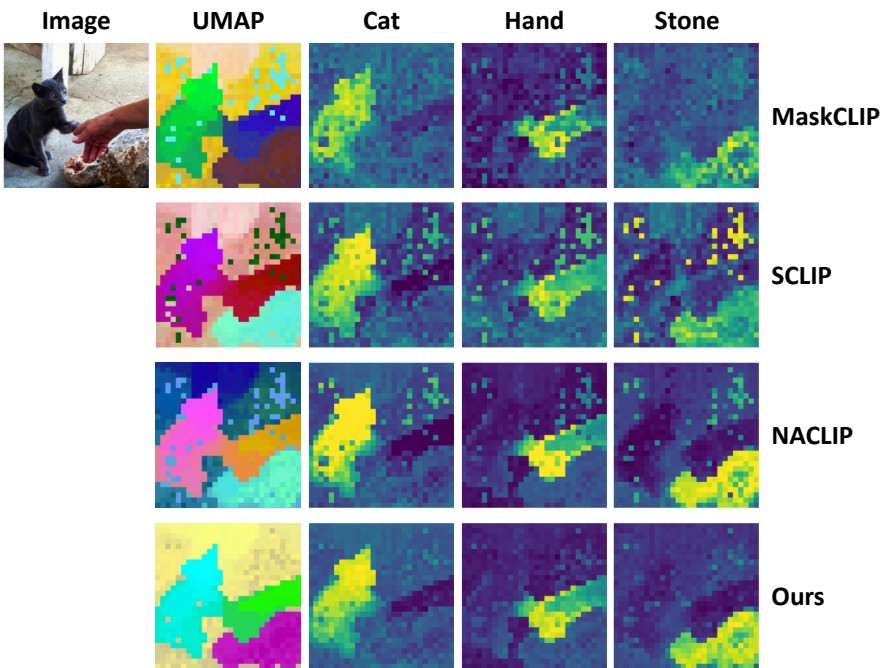

Figure S.4: **Zero-shot Heatmap Results.** Our results have fewer artifacts than other methods.

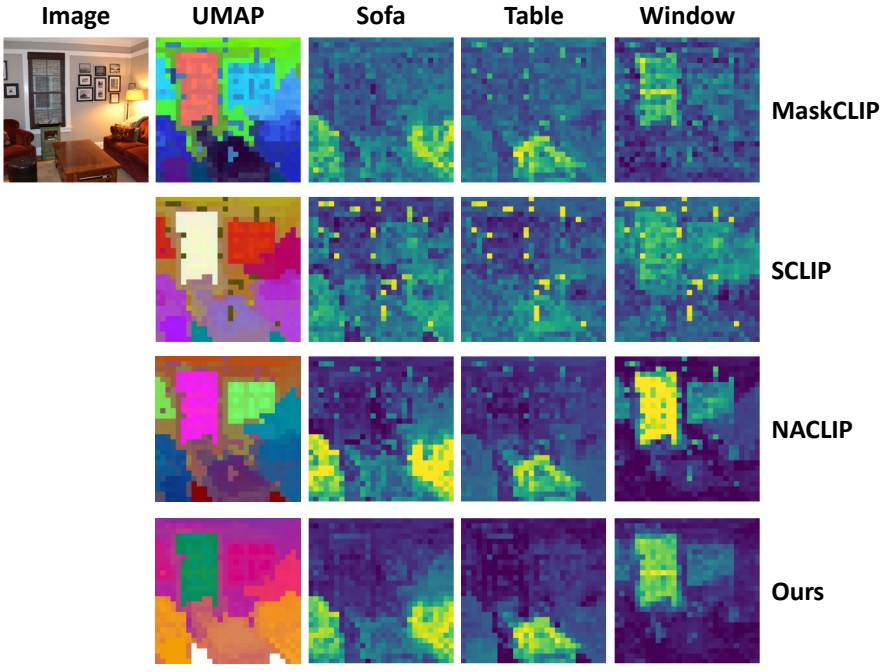

Figure S.5: **Zero-shot Heatmap Results.** Our results have fewer artifacts than other methods.

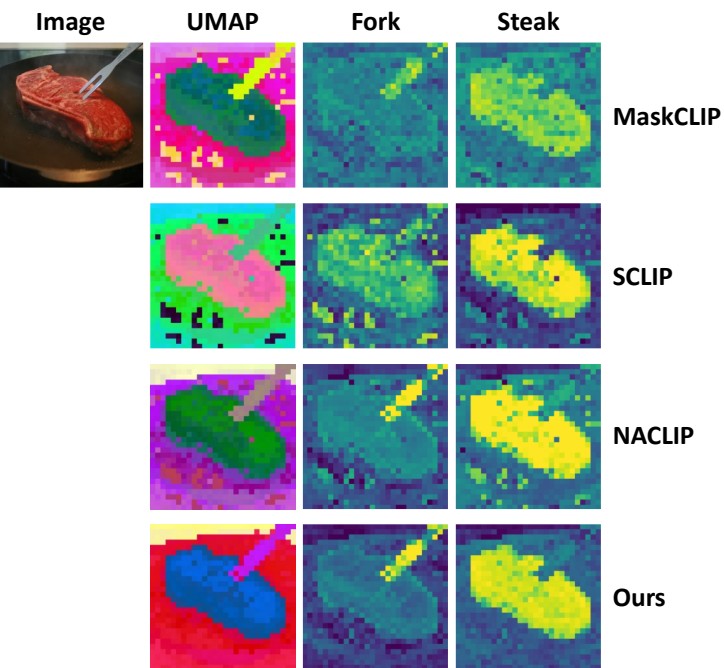

Figure S.6: **Zero-shot Heatmap Results.** Our results have fewer artifacts than other methods.

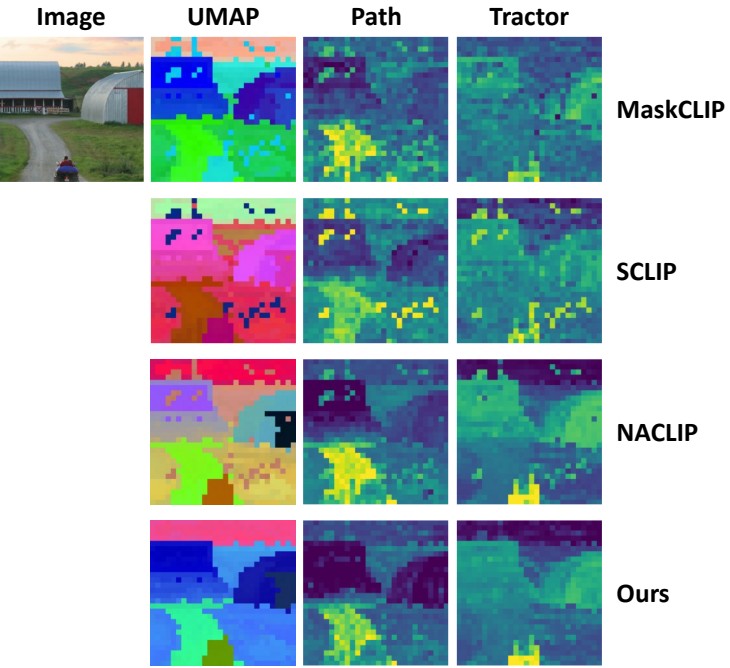

Figure S.7: **Zero-shot Heatmap Results.** Our results have fewer artifacts than other methods.

# G    Optimal Feature Aggregation

Let $f_1, \ldots, f_n \in \mathbb{R}^d$ be feature vectors of a single patch from $n$ different transformations of an input image $\mathcal{I}$. We seek the optimal aggregated feature $f^*$ that minimizes the total squared error:

$$f^* = \arg\min_f \sum_{i=1}^{n} \|f_i - f\|_2^2 \tag{1}$$

Expanding the objective gives us:

$$\arg\min_f \sum_{i=1}^{n} (f_i^\top f_i - 2f_i^\top f + f^\top f) \tag{2}$$

Dropping constant terms that do not change the optimum:

$$= nf^\top f - 2 \left( \sum_{i=1}^{n} f_i^\top \right) f \tag{3}$$

Dividing and multiplying the right side by $n$:

$$= nf^\top f - 2n \left( \sum_{i=1}^{n} \frac{1}{n} f_i^\top \right) f \tag{4}$$

Dividing the equation by $n$ as whole shows us that we need to minimize:

$$\|f - \frac{1}{n} \sum_{i=1}^{n} f_i\|_2^2 \tag{5}$$

So it can be derived that the mean of the feature vectors is the minimizer under MSE loss:

$$f^* = \frac{1}{n} \sum_{i=1}^{n} f_i \tag{6}$$

