# OpenReview forum: "Vision Transformers with Self-Distilled Registers"
_NeurIPS.cc/2025/Conference — NeurIPS 2025 spotlight_

### Official Review · Reviewer_EcZk · 2025-06-21

**Clarity:** 3
**Significance:** 3
**Originality:** 3
**Rating:** 5
**Confidence:** 4

**Summary:**

This paper introduces an interesting approach for learning registers in Vision Transformers (ViTs), building upon recent findings that additional registers help remove artifacts captured by the class token. The authors propose a novel method to clean artifact-contaminated feature maps using image augmentation, then leverage knowledge distillation to learn additional registers from the cleaned feature maps without requiring full training from scratch. Experiments on multiple segmentation tasks demonstrate the efficiency of the proposed method.

**Questions:**

All baseline comparisons appear to use models fine-tuned from pre-trained CLIP. How does the proposed method compare against the original register-ViT trained from scratch?

Would it be possible to include experiments comparing against models trained from scratch without registers? This would help isolate the contribution of the distillation approach versus the underlying register mechanism.

Given that data augmentation eliminates artifacts from feature maps during training, does this suggest that test-time data augmentation would also improve task performance? It would be valuable to design experiments that isolate and quantify the specific contribution of the data augmentation component. For instance, could the authors compare: (1) models trained with their full pipeline, (2) models trained with distillation but without the augmentation-based artifact cleaning, and (3) models that apply similar augmentation strategies at test time? Such ablations would help disentangle whether performance gains stem primarily from the cleaner training signal or from the augmentation strategy itself.

**Ethical Concerns:**

["NO or VERY MINOR ethics concerns only"]

**Final Justification:**

Following the rebuttal, the authors have demonstrated a clear commitment to addressing the raised concerns and have outlined concrete plans for revision.

Based on these improvements and the authors' commitment to implementing the promised changes, I have revised my evaluation score upward. I trust the authors will follow through with their proposed revisions in the final manuscript.

**Limitations:**

The limitation has been well discussed.

**Quality:**

3

**Strengths And Weaknesses:**

Strengths:

- The motivation is well-founded and addresses a relevant problem in the recent popular finding on ViT with learnable registers. The paper seeks to improve upon recent advances in register tokens by developing an efficient approach to learn these registers without costly full training or fine-tuning procedures.

- The proposed artifact elimination method using feature map augmentation is simple, novel, and efficient. The approach offers a practical solution to a known issue in ViT architectures.

- The distillation-based learning framework is well-motivated and demonstrates strong empirical performance according to the experimental results, providing an effective alternative to more expensive training procedures.



Weaknesses:

I didn't find major weakness. Here I list some flaws that should further improve the paper.

- The methodology section would benefit from more detailed mathematical formulation. The authors should include explicit equations describing how the final cleaned feature maps are computed.

- Section 3.3 lacks sufficient detail regarding the distillation methodology. The paper should provide clearer justification for the chosen loss function and include ablation studies demonstrating why this particular loss formulation was selected over alternatives.

- While computational efficiency is claimed as a key advantage, the paper lacks quantitative comparisons of training time, memory usage, and computational overhead between the proposed method and full training/fine-tuning approaches with registers.

There are also some questions that I would like to discuss with authors. Please refer the below section.

---

> ### Author Rebuttal · Authors · 2025-07-30
>
> We appreciate your detailed and concrete suggestions! We will incorporate all of your feedback into our paper.
>
> > **Q1) Detailed method description**
>
> We agree that the paper could be improved with additional details about our distillation procedure. We have already incorporated your suggestions into our internal revision. Due to NeurIPS OpenReview policy this year, we cannot upload a revision.
>
> We provide a condensed pseudocode here, where we assume we are targeting a vision transformer with patch size $k \times k$, an image $\mathcal{I}$ which is an integer multiple of the patch size. Augmentation parameters $(\theta_1, \theta_2, ..., \theta_n)$ for up to $n$ augmentations are generated randomly from a constrained distribution, and each augmentation consists of a binary horizontal flip parameter $\text{flip}_i \in \{0,1\}$, and a vertical/horizontal offset tuple $(x_i, y_i)$. We also have a tuple grid of 2D image coordinates $C$, which helps us index the locations in an image. An augmentation function takes as input $(\mathcal{I}, C, \theta_i)$ and generates the corresponding shifted/flipped image and coordinates.
>
> Pseudo code:
> > Inputs: Image $\mathcal{I}\in\mathbb{R}^{H\times W \times 3}$; Image space coordinates $C\in[0,1]\times[0,1]$; Augmentation parameters $\theta_{1}, \theta_2, ..., \theta_n$; Augmentation function $T$; ViT teacher model $f_\text{teacher}$; Zero-init feature accumulator tensor $Q$; Zero-init count occurrence tensor $K$
>
> > For i in ${1,...,n}$
> >> $\theta_i = (x_i, y_i, \text{flip}_i)$
> >>
> >> ($\mathcal{I}_i, C_i$) = $T(\mathcal{I}, C, \theta_i)$ # Augment image, and generate the image-coordinates of the shifted/flipped image
> >>
> >> Dense Feature $F_i = f_\text{teacher}(\mathcal{I}_i)$
> >>
> >>$(F^\text{valid}_i , C^\text{valid}_i) = T^{-1}(F_i , C_i, \theta_i)$ # Given the coordinates and the feature map of the augmented image, we map the feature back to the position in the original image
> >>
> >> $Q[C^\text{valid}_i] = Q[C^\text{valid}_i] + F^\text{valid}_i$
> >>
> >> $K[C^\text{valid}_i] =K[C^\text{valid}_i] + 1$
> >>
> >> return $Q/K$
>
> We include a proof on page 12 of the supplemental showing that this achieves the same expected value, as gradient-based MSE optimization used in DVT neural field learning.
>
> **We will include additional details in our updated revision.**
>
> > **Q2) Ablation study of the distillation loss**
>
> This is a good suggestion. Here we perform an ablation study using MSE, cosine similarity, and both combined (used in our work, and also used in DVT) -- in this table higher is better:
>
>
> | Loss                     |   cos sim   |   MSE   | MSE & cos sim |
> |--------------------------|:------:|:------:|:------------:|
> | Median Cosine Similarity | 0.9652 | 0.9673 |  **0.9720**  |
>
> Note that cosine similarity alone does not preserve magnitude, and occasionally yields training instability when used alone.
>
> **We will include this result in the revision.**
>
> > **Q3) Computational efficiency**
>
> We agree that additional quantification of training and inference costs is beneficial. Given that custom kernels can significantly influence both performance and memory usage, our evaluation focuses on model size, FLOPs, and measured wall-clock training time. To ensure a consistent baseline, training time was measured using the official DVT repository code without modifications. Since the original DVT implementation specifies a single GPU for its feature extraction and learning phases, we similarly constrained all stages of our model to a single GPU for this comparison. The evaluated models (DVT and PH-Reg) are based on the same architecture: DINOv2 ViT-B.
>
> Training wall-clock time:
>
> | Method | Stage 1 Extraction | Stage 1 Distillation | Stage 2 Training  | Total               |
> |--------|--------------------|----------------------|-------------------|---------------------|
> | DVT    | 2998 min           | 18340 min            |  570 min            | 21908 min (365.1 h) |
> | PH-REg | -                  | -                    |        -           | 9000 min (150 h)    |
>
> During training, DVT must store serialized Instant-NGP features, requiring **approximately 1.4TB of storage**. In contrast, our method imposes no supplementary storage requirements.
>
> To assess testing costs fairly, all models were evaluated with parameters cast to `fp32` and using an `eager` attention implementation. We account for positional embeddings adjusted for 448x448 resolution into the PH-Reg parameter count. Regarding MaskCLIP and NACLIP, their official implementations currently retain the same parameter count as the original CLIP model, though MaskCLIP can achieve a minor reduction by removing the final query and key head.
>
> In the case of DVT, we assessed their second-stage model, which pairs a transformer block denoiser with an original vision transformer.
>
> | Method                                | GFLOPs | Params (M) |
> |---------------------------------------|:------:|:----------:|
> | MaskCLIP                              |  62.89 |    86.19   |
> | NACLIP                                |  64.76 |    86.19   |
> | NACLIP + test time augmentation (10x) |  647.6 |    86.19   |
> | NACLIP + DVT                            |  70.32 |    94.07   |
> | CLIP + PH-Reg (Ours)                  |  64.16 |    86.66   |
>
> We achieve notable efficiency gains over DVT: a 2x reduction in training time, complete elimination of training storage costs (0 TB vs. 1.4 TB), and ~10% savings in both inference FLOPs and model parameters. **These metrics will be added in the upcoming revision of our paper.**
>
>
> > **Q4) Comparison against the official DINOv2 + registers**
>
> Here we perform a comparison against DINOv2 + registers, using the official weights provided by facebook research:
>
>
> |         Method         |   VOC21   |   VOC21   |   ADE20k  |   ADE20k  |    NYUv2   |    NYUv2   |     NYUv2     |
> |:----------------------:|:---------:|:---------:|:---------:|:---------:|:----------:|:----------:|:-------------:|
> |                        |  mIoU(↑)  |  mAcc(↑)  |  mIoU(↑)  |  mAcc(↑)  |   RMSE(↓)  | Abs Rel(↓) | $\delta_1$(↑) |
> |         DINOv2         |   84.13   |     92    |   47.82   |    60.5   |   0.4566   |   0.1391   |     82.92     |
> |   DINOv2 + registers   |    83.9   |   91.04   | **48.75** | **62.48** |   0.4382   |   0.1295   |     85.13     |
> | DINOv2 + PH-Reg (Ours) | **84.85** | **92.46** |   48.66   |   61.57   | **0.4306** | **0.1216** |   **86.35**   |
>
> We find that our method achieves consistent improvements all on metrics compared to the original DINOv2 (without registers). Our method has better performance on VOC21 segmentation and NYUv2 depth prediction compared to `DINOv2 + registers`, while having slightly inferior performance on ADE20k. Note that the VOC21 mAcc degradation of `DINOv2 + registers` compared to `DINOv2` was also observed by DVT. For VOC21 mIoU, DVT observes a very small improvement, while we observe a very small degradation. For fairness, we ran our own evaluation using the same resolution for all methods -- and the evaluation settings are slightly different than those used DVT in their paper.
>
> Given that our method can be completed using around 1.5 days on 4 Nvidia Ada GPUs, we believe our framework offers a very attractive alternative to full model retraining (estimated at several GPU-months if referencing OpenCLIP ViT-B numbers). **We will include this comparison in the revised paper.**
>
> > **Q5) Ablation studies of distillation approach**
>
> Here we evaluate the contribution of each component in OpenAI CLIP + PH-Reg  (mIoU, higher is better):
>
> | Approach                     |   VOC21   | Context60 | COCO object |   VOC20   | Context59 | COCO stuff | Cityscape |   ADE20k  |    Avg    |
> |------------------------------|:---------:|:---------:|:-----------:|:---------:|:---------:|:----------:|:---------:|:---------:|:---------:|
> | Vanilla                      |   49.27   |   25.46   |    26.94    |   66.56   |   28.62   |    18.80   |   28.33   |   13.70   |   32.21   |
> | TTA only (10x augmentation)                          |   51.41   |   28.13   |    29.00    |   69.58   |   31.03   |    20.25   |   31.82   |   15.20   |   34.55   |
> | Distill from teacher w/o registers w/o TTA              |   61.16   |   33.51   |    34.51    |   81.51   |   36.70   |    23.96   |   35.74   |   18.34   |   40.68   |
> | Distill from teacher w/ registers  w/o TTA                |   61.27   |   33.52   |    34.39    |   81.52   |   36.74   |    23.92   |   35.55   |   18.38   |   40.66   |
> | Distill from teacher w/o reg + TTA                |   62.48   |   34.28   |    35.00    |   82.27   |   37.62   |    24.46   |   36.83   |   18.92   |   41.48   |
> | Full Pipeline (w/ reg + TTA) | **63.01** | **34.52** |  **35.27**  | **83.05** | **37.88** |  **24.66** | **37.17** | **19.22** | **41.85** |
>
> By comparing our full pipeline with the `Distill from teacher w/o registers w/o TTA` baseline, we find that approximately half the improvement comes from registers, and half comes from test-time augmentation of the teacher network.
>
> **We will include these results in the revision.**
>
> > **Conclusion**
>
> We are very grateful for the highly detailed and actionable suggestions. We have included additional details for our method, additional experiments on distillation loss, method efficiency, comparison against models with registers trained from scratch, and ablation studies of all components of our pipeline.
>
> Given our experiments and clarifications, we hope we have resolved your questions! We look forward to any additional suggestions you may have.

---

### Official Review · Reviewer_MibD · 2025-07-01

**Clarity:** 4
**Significance:** 3
**Originality:** 3
**Rating:** 5
**Confidence:** 4

**Summary:**

The paper continues the recent area of research that indicates the presence of artifacts in vision transformers, and the introduction of register tokens to remedy this. The initial idea introduced and trained register tokens in the main architecture and discarded them for the loss and gradient computation. This paper introduces an important modification from there --  self-distilled registers which can be added to a pretrained model, and leads to good results at minimal retraining cost.

**Questions:**

Please see the strengths and weaknesses section above.

**Ethical Concerns:**

["NO or VERY MINOR ethics concerns only"]

**Final Justification:**

The presented work makes a significant stride on the concept of artifacts in vision transformers and how to mitigate them. Based on my earlier review, as well as the rebuttal, I believe this work would be relevant for further research directions, and thus I will continue to maintain my score (Accept).

**Limitations:**

yes

**Quality:**

4

**Strengths And Weaknesses:**

Strengths:

1. The paper presents a strong idea that significantly reduces the training costs, while also solving the issue of artifacts using the same registers idea.
2. The experiment set and ablations are quite detailed and cover a variety of tasks, including semantic segmentation, depth estimation. The ablations cover various scenarios, like varying the number of register tokens, unlocking various layers, as well as augmentation settings during test-time adaptation.

Weaknesses and Questions:

1. As the authors point out, the method currently lacks the ability to determine artifacts in a dynamic manner. While this would require further research, a small discussion on potential future directions would be helpful.
2. The paper "Vision Transformers Need Registers" suggested that the artifact tokens are high norm while this paper suggests that this isnt always the case. Can this be inspected further to understand if this inconsistency is the result of small architectural modifications? If tried out already, do the authors find a model trained along with registers from scratch to be consistent to the mentioned paper? Another recent paper "Vision Transformers Don't Need Trained Registers" uses the high norm claim to build a method (this is purely a mention and does not affect my rating as it is concurrent work)
3. The "Vision Transformers Need Registers" paper suggests that the artifact tokens tend to contain global rather than local context, typically class information, etc. Do the authors find this observation while distilling register tokens as well?

---

> ### Author Rebuttal · Authors · 2025-07-30
>
> We are deeply grateful for your helpful comments! We will address specific questions below, and will include additional details in a revision.
>
> > **Q1) Dynamic determination of artifact tokens**
>
> This is a very interesting idea. Our current framework only removes the artifact tokens, but identifying the artifact tokens dynamically could be a promising direction. In principle, a "corrector" network could  be learned by estimating the difference between the original "raw" dense features and denoised teacher features. This network could be conditioned on the original dense features and learn the artifacts.
>
> After identification, the artifacts could be either removed or shifted to a register using ideas from recent work.
>
> **We will include a discussion on this in the upcoming revision.**
>
> > **Q2) Magnitude of Artifact Tokens**
>
> We were not previously aware of `Vision Transformers Don't Need Trained Registers`. After a careful reading, we agree it is highly relevant and will include a reference to this work in our revision. It is striking that artifact tokens could be removed using heuristics alone.
>
> We would like to clarify that for CLIP models, we focus on the zero-shot open-vocabulary segmentation task. In this case, using the `value features` (or some modified convex combination of the value features across the image) of the last attention block are the preferred approach [1, 2, 3, 4].
>
> As we show in `Figure 6`, we find when value features (or convex combination of value features) from the last layer are examined for CLIP models, indeed the artifact tokens are primarily low magnitude, rather than high magnitude in OpenAI's CLIP and OpenCLIP. However in DFN-CLIP, which shares the same architecture and training objective as the OpenCLIP model we evaluate -- the artifact magnitudes are more mixed. The primary difference between DFN-CLIP and OpenCLIP are the training images. This suggests that training data may play an important role in the emergence of artifact tokens.
>
> Discussions with some DINOv2 authors offline, also suggests data may be a factor. When discussing why artifacts appear in DINOv2 but not DINOv1, one DINOv2 author suggests that increase proportion of images with white backgrounds (collected from online retail) in the DINOv2 training set may be the cause, whereas DINOv1 was trained with a higher ratio of naturalistic photographs. While this is plausible, we do not believe it is a complete explanation.
>
> We will include additional discussion and a citation to the `Don't Need Trained Registers` work in our paper revision.
>
> [1] `Extract Free Dense Labels from CLIP (ECCV 2022)`
>
> [2] `SCLIP: Rethinking Self-Attention for Dense Vision-Language Inference (ECCV 2024)`
>
> [3] `ClearCLIP: Decomposing CLIP Representations for Dense Vision-Language Inference (ECCV 2024)`
>
> [4] `Pay Attention to Your Neighbours: Training-Free Open-Vocabulary Semantic Segmentation (WACV 2025)`
>
> > **Q3) Global information in artifact and register tokens**
>
> It is fascinating that there seems to be a diverse set of views on the nature of artifacts -- [5] argues they include global information, while [6] argues they are image independent, with very recent work by `Lappe and Giese` [7] finding that some dimensions of artifact tokens contain global information -- with results varying due to model size and layernorm.
>
> Similarly, there are disagreements about what causes artifact tokens to occur. DVT suggests that the positional embeddings are primarily responsible for artifacts. However in `section 4.3`, we find that artifacts cannot be fully resolved by finetuning the positional embeddings alone.
>
> We have performed a visualization of the register tokens learned using PH-Reg -- which we will include in an upcoming revision. We find that in our current setup, the attention maps for the registers do indeed correspond to semantic objects within the image. However qualitatively the effect seems weaker than the models trained from scratch. One potential reason is that we do not constrain the `[CLS]` token of the student to decode global image information, and instead focus on the quality of dense representations.
>
> We will include this attention map visualization in the upcoming revision, and discuss why our results may differ from models trained with `[CLS]` token constraints.
>
> [5] `Vision Transformers Need Registers (ICLR 2024)`
>
> [6] `SINDER: Repairing the Singular Defects of DINOv2 (ECCV 2024)`
>
> [7] `Register and CLS tokens yield a decoupling of local and global features in large ViTs (Arxiv)`
>
> > **Conclusion**
>
> We thank you for your comments, and we hope that this clarifies our results! We will update the paper to reflect your suggestions.

---

> > ### Comment · Reviewer_MibD · 2025-08-04
> >
> > I thank the authors for their rebuttal, and will continue to maintain my score as Accept, thanks for the good work!

---

> > > ### Author Response · Authors · 2025-08-04
> > >
> > > We are deeply grateful for the positive assessment you've given to our paper! Thank you again for your suggestions and comments.

---

### Official Review · Reviewer_cHXN · 2025-07-03

**Clarity:** 4
**Significance:** 3
**Originality:** 4
**Rating:** 5
**Confidence:** 3

**Summary:**

Recent research shows that big Vision Transformer models such as DINOv2 or OpenCLIP contain artifact tokens that encode undesired information. While this information could be relevant, it does not provide the local semantics that should represent the patch token, degrading the performance of the ViT on tasks that require fine-grained patch information. Authors propose Post Hoc Registers (PH-Reg), an elegant solution to the problem that, contrary to the current SoTA, does not require full re-training or additional labelled data. Main contributions can be summarised in their simple but effective self-distillation training and the artifact denoising strategy that enables the whole training by acting as a training target.

**Questions:**

From the weaknesses explained before:
1. Can authors provide more in-depth analysis regarding the performance gain decay as the number of registers increases?
2. Is the proposed method more efficient than DVT? By how much in terms of memory and time?
3. Improvements seem to be higher on VLMs, can the authors provide more in-depth insights about this behaviour?

**Ethical Concerns:**

["NO or VERY MINOR ethics concerns only"]

**Final Justification:**

The author's answers have addressed my concerns; therefore, I keep my rating and recommend the acceptance of this paper.

**Limitations:**

Yes

**Quality:**

3

**Strengths And Weaknesses:**

**Strengths:**

Technically sound method that successfully addresses the identified problem.

PH-Reg stands as a model "corrector", automatically improving any ViT-based SoTA vision model that suffers from explained artifacts.

Provided quantitative and qualitative analysis clearly display the benefits of the proposed method, highlighting especially the qualitative results.

I find the denoising process intriguing. While a corrected method would always be more efficient, the described denoising strategy already manages to remove most of the artifacts without any specific training.

The authors yield several insights about the artifacts that may help the rest of the community working in the same field.

The paper is well-written and correctly organised.



**Weaknesses:**

The authors claim that performance gain greatly decays after increasing the number of registers past 4/8. I would like to see more analysis on this, does this behaviour depend on the pretrained model? If not, what are the insights that could justify this general behaviour?

Efficiency analysis regarding required memory (which I assume is low) and time between DVT and proposed approach would be interesting and could further prove the superiority of PH-Reg if they end up being much more efficient than DVT.

Proposed method seems to be much better when applied to VLMs, more in-depth insights about this behaviour would be appreciated.

Being a SoTA model "corrector", I believe the proposed approach has a strong short-term applicability. However, Registers are being widely applied already, and, at some point, models won't require a "corrector" anymore, dramatically decreasing the applicability of the proposed work.

---

> ### Author Rebuttal · Authors · 2025-07-30
>
> Thank you for your detailed suggestions! We will incorporate all of your feedback into our paper.
>
>
> > **Q1) Investigation of register counts across models**
>
> This is an interesting question. In the original paper we evaluate the number of registers only on the OpenAI CLIP ViT model. Based on your suggestion, we further evaluate OpenCLIP, DFN-CLIP, and DINOv2 -- three models that were trained using different datasets (in the case of OpenCLIP and DFN-CLIP) and different objectives (DINOv2). In all cases we observe decreased rates of performance gain when additional registers are added, across median cosine similarity (higher is better):
>
> | Number of Registers |    1   |    2   |    4   |    8   |     16     |
> |---------------------|:------:|:------:|:------:|:------:|:----------:|
> | OpenCLIP            | 0.7852 | 0.7858 | 0.7985 | 0.8025 | **0.8139** |
> | DFNCLIP             | 0.7553 | 0.7581 | 0.7687 | 0.7751 | **0.7854** |
> | DINOv2              | 0.9265 | 0.9260 | 0.9327 | 0.9346 | **0.9350** |
>
> We are not the first to observe this, as similar trends are observed in the saturation of dense task performance in the original "Vision Transformers Need Registers" [1] paper. Both our own visualizations (will be added to upcoming revision), and those in [1] suggest that registers learn to focus on semantic objects within the image. We speculate that the saturation in performance is due to the limited number of objects in common images.
>
> One potential follow-up experiment could be based on synthetic datasets like CLEVR, where the number of objects within each image is known. These datasets could be used to investigate if the performance benefit of register tokens correlates with the average number objects across a dataset.
>
> **We will include the table for different models in an upcoming revision of our paper.**
>
> > **Q2) Efficiency of PH-Reg compared to DVT**
>
> We agree that additional quantification regarding both training and inference cost would be useful. Due to the potential influence of custom kernels on both performance and model memory usage, here we evaluate model size, FLOPs, and wall clock training time. When evaluating training time, we utilize the official code provided in the DVT repository without any modification. The original DVT code specifies a single GPU for feature extraction and learning -- for fairness we also limit all stages of our own model to a single GPU here. For DVT and PH-Reg, we evaluate the same model (DINOv2 ViT-B) at the same image resolution.
>
> Training wall-clock time:
>
> | Method | Stage 1 Extraction | Stage 1 Distillation | Stage 2 Training  | Total               |
> |--------|--------------------|----------------------|-------------------|---------------------|
> | DVT    | 2998 min           | 18340 min            |  570 min            | 21908 min (365.1 h) |
> | PH-REg | -                  | -                    |        -           | 9000 min (150 h)    |
>
> During training, DVT also requires storage of the serialized Instant-NGP features. Which **requires approximately 1.4TB**, while our method requires no additional storage cost during training.
>
> When evaluating testing cost for all models, we cast parameters to `fp32` dtype, and use `eager` attention implementation for all models. For our method, we include the positional embeddings adapted for 448 resolution. For MaskCLIP and NACLIP, current official implementations have the same number of parameters as the original CLIP, although a small reduction can be achieve in MaskCLIP by discarding the last q,k head.
>
> For DVT, we evaluate their stage 2 model, corresponding to the transformer block denoiser coupled to a original vision transformer.
>
> | Method                                | GFLOPs | Params (M) |
> |---------------------------------------|:------:|:----------:|
> | MaskCLIP                              |  62.89 |    86.19   |
> | NACLIP                                |  64.76 |    86.19   |
> | NACLIP + test time augmentation (10x) |  647.6 |    86.19   |
> | NACLIP + DVT                            |  70.32 |    94.07   |
> | CLIP + PH-Reg (Ours w/ 16 registers)                  |  64.16 |    86.66   |
>
> Our method is significantly faster to train (2x speed), and utilizes significantly less storage during training compared to DVT (0 vs 1.4TB). Our method also utilizes approximately 10% fewer FLOPs and 10% fewer parameters during inference compared to DVT. **We will include these details in an upcoming revision of our paper.**
>
> > **Q3) Why so well on Vision Language Models?**
>
> Indeed, our performance gains are the most striking on models trained with an image-language contrastive loss, where OpenAI CLIP, OpenCLIP, and DFN-CLIP were trained on different datasets. While we observe performance improvements on DINOv2 compared to the version without PH-Reg, the improvements are not as striking.
>
> We show in Figure 6, for CLIP family models the artifacts primarily manifest as lower magnitude tokens when utilized for open-vocabulary segmentation. While in DINOv2 the artifacts are high magnitude -- confirming the observation in "Vision Transformers Need Registers" [1].
>
> We are unsure what causes the difference in magnitudes of artifact tokens, and our offline discussions with DINOv2 authors suggests that they are also unsure why artifacts appear in DINOv2 but not DINOv1. One co-author of DINOv2 suggested that the increase in single-color background images (collected from online retail), and the decrease in natural photographs in the training dataset was a likely reason in the artifact tokens.
>
> Regardless of the cause, if artifact tokens are low magnitude, then their influence decreases after averaging. We believe this is the reason of the increased performance of PH-Reg in CLIP family models. For future work, it may make sense to decouple patch direction and magnitude prior to distillation.
>
> We will include a discussion of this in the revision.
>
> `[1] Vision Transformers Need Registers (ICLR 2024)`
>
> > **Conclusion**
>
> We thank you for providing detailed and thoughtful feedback. Following your suggestions, we have run an evaluation to measure the impact of register counts in different models, the training efficiency of our method versus DVT, and the inference computational cost. We hope our answers have helped clarify your questions, and look forward to any additional discussion.

---

> > ### Comment · Reviewer_cHXN · 2025-08-02
> >
> > Thanks for taking into consideration my comments and providing new figures to answer my questions.
> >
> > Your answers have addressed my concerns, and therefore, I keep my rating and recommend the acceptance of this paper.

---

> ### Author Response · Authors · 2025-08-04
>
> We are very grateful for the positive assessment you've given to our work!
>
> We would like to again express our appreciation for your suggestions.
>
> As a gentle reminder, for NeurIPS this year -- you may have to edit your review with a "Final Justification". Thank you for taking the time!

---

### Official Review · Reviewer_H7Es · 2025-07-03

**Clarity:** 3
**Significance:** 3
**Originality:** 2
**Rating:** 4
**Confidence:** 4

**Summary:**

The paper introduces Post Hoc Registers (PH-Reg), an efficient self-distillation framework to mitigate artifact tokens in pre-trained ViTs without full retraining or labeled data. By initializing teacher and student networks from the same pre-trained ViT, freezing the teacher, and augmenting its inputs(e.g.. random offsets, flips), PH-Reg generates denoised dense embeddings to guide student optimization, only updating a small subset of weights including randomly initialized register tokens. This approach effectively reduces artifact tokens, improving zero-shot and linear-probe performance in segmentation and depth prediction.Key contributions include an efficient test-time augmentation denoising scheme, selective finetuning of student components for clean feature learning, and demonstrated enhancements in dense representation consistency for fine-grained spatial tasks, as validated across multiple benchmarks.

**Questions:**

1.	The denoise step needs to ablate the shift ratio. And i'm wondering why not vertical fllp?

2.	It seems all the experiments done with salient object. What about the results on more complex scenarios like coco and lvis?

See weaknesses for other questions.

If the author can address the above issues and provide a clearer theoretical analysis to demonstrate the effectiveness and robustness of the proposed method, I would be willing to raise the score.

**Ethical Concerns:**

["NO or VERY MINOR ethics concerns only"]

**Final Justification:**

The author's answers have addressed most of my concerns. Meanwhile, considering other reviewers' comments, I lean to accept the paper.

**Limitations:**

It would be better to discuss the failure cases of Dino to analyze the potential shortcomings.

**Quality:**

2

**Strengths And Weaknesses:**

Strengths:

1.	PH-Reg introduces a novel self-distillation framework that integrates register tokens into pre-trained ViTs without full retraining, leveraging test-time augmentation to denoise dense features as the teacher.This approach avoids the computational cost of training from scratch while effectively reducing artifact tokens that degrade fine-grained localization tasks.

2.	PH-Reg demonstrates consistent improvements across diverse dense prediction tasks (e.g., semantic segmentation, depth estimation) under zero-shot and linear probing setups, outperforming state-of- the-art baselines on multiple benchmarks.

Weaknesses:

1.	Denoising dense predictions in a self-supervised manner is a common strategy in image processing works. The paper simply adopts the idea and finds that it also helps with the denoise at feature level. However, it lacks formal theoretical justification for the effectiveness of register tokens and test-time augmentation in artifact mitigation, focusing primarily on empirical validation.

2.	While PH-Reg excels on CLIP-based models, it underperforms DVT on DINOv2 due to DVT's static artifact estimator, indicating sensitivity to model-specific artifact characteristics.

3.	The denoising efficacy relies on carefully hand-designed augmentations (e.g,random offsets), which may require task-specific tuning for optimal results. It's also needed to ablate the shift ratio impact.

---

> ### Author Rebuttal · Authors · 2025-07-30
>
> We appreciate your excellent suggestions! Below are our responses to specific questions. We look forward to further discussion, and welcome additional comments.
>
> > **Q1) Why not vertical flips as augmentation?**
>
> We avoid vertical flips in our augmentation pipeline because this violates real-world physical constraints. Gravity imposes a consistent vertical orientation in natural scenes (e.g., sky above ground, human faces upright), making vertically flipped images unrealistic. In contrast, horizontal flips preserve vertical orientation while simulating naturally occurring viewpoint variations (e.g., an object facing left/right). Augmentations like horizontal flips and shifts expand the training distribution with plausible variations.
>
> The mmseg package [1] commonly used for dense vision evaluation, similarly defaults to `horizontal` for `RandomFlip`. Generally other works in computer vision also avoid using vertical flips as augmentation. For example, the ResNet paper [2] and DeiT III [3] training recipe (utilized by DINOv2 for ImageNet) adopt horizontal flips without vertical flipping in their augmentation pipeline.
>
> We will update our revision to include a paragraph clarifying why we do not utilize vertical flips in our augmentation function.
>
> [1] https://mmcv.readthedocs.io/en/2.x/api/generated/mmcv.transforms.RandomFlip.html
>
> [2] `Deep Residual Learning for Image Recognition (CVPR 2016)`
>
> [3] `DeiT III: Revenge of the ViT (ECCV 2022)`
>
> > **Q2) Ablation over shift ratios**
>
> This is a great question. The optimal shifting ratio of 15% was determined using extensive experiments. Here we provide a table where additional shift ratios are explored using NACLIP teacher features (note that only the student network has registers).
>
> We find that a shifting ratio of 15% provides the optimal performance (mIoU, higher is better):
>
> | ratio |   VOC21   | Context60 | COCO object |   VOC20   | Context59 | COCO stuff | Cityscape |  ADE20k  |    Avg    |
> |-------|:---------:|:---------:|:-----------:|:---------:|:---------:|:----------:|:---------:|:--------:|:---------:|
> | 0%    |   58.88   |    32.20   |    33.15    |    79.70   |   35.16   |    23.30    |   35.48   |   17.42  |   39.41   |
> | 10%   | **60.49** | **32.91** |  **33.73**  | **80.47** | **35.94** |    23.86   |    36.60   |   17.94  |   40.24   |
> | 15%   |   60.47   |   32.89   |    33.61    |   80.36   | **35.94** |  **23.89** |   36.87   | **18.10** | **40.27** |
> | 20%   |   60.39   |   32.82   |    33.46    |   79.93   |   35.85   |    23.86   |   36.98   |   18.07  |   40.17   |
> | 25%   |   60.26   |   32.75   |    33.26    |   79.75   |   35.75   |    23.78   |   37.02   |    18.00    |   40.07   |
> | 30%   |   60.14   |   32.76   |    33.25    |   79.39   |   35.77   |    23.77   | **37.07** | **18.10**   |   40.03   |
>
> ***We will add this table to the paper in an upcoming revision.***
>
> > **Q3) Number of registers**
>
> We find that the rate of improvement decreases as additional registers are added. This trend is observed over all models. We provide additional results here (median cosine similarity, higher is better), and will include these results in an upcoming revision:
>
> | Number of Registers  | 1      | 2      | 4      | 8      | 16     |
> |----------|--------|--------|--------|--------|--------|
> | OpenCLIP | 0.7852 | 0.7858 | 0.7985 | 0.8025 | **0.8139**|
> | DFNCLIP  | 0.7553 | 0.7581 | 0.7687 | 0.7751 | **0.7854** |
> | DINOv2   | 0.9265 | 0.9260 | 0.9327 | 0.9346 | **0.9350** |
>
> > **Q4) Proof of convergence**
>
> We would like to clarify, that a key advantage of our method over DVT is that we avoid expensive gradient-based neural field optimization. On `page 12` we include a proof showing that we obtain the same optimal result in expectation as the neural field stage of DVT.
>
> Let $f_1, ..., f_n \in R^d$ be feature vectors of a single patch from $n$ transformations of an input image. DVT optimizes a neural field using MSE loss gradients, while we take the unweighted mean. We summarize the proof here:
>
> Let $f^{*} = \text{argmin}_f \sum_i^n ||f_i-f||^2_2$
>
> `Expanding the objective gives us`: $\text{argmin}_f \sum_i^n (f_i^{T}f_i - 2f_i^{T}f + f^{T}f)$
>
> `Dropping constant terms that do not change the optimum`: $n f^{T}f - 2(\sum_i^nf_{i}^{T})f$
>
> `Dividing and multiplying the right side by n`: $n f^{T}f - 2n(\sum_i^n\frac{1}{n}f_{i}^{T})f$
>
> `Dividing the equation by constant n shows that we obtain the same optimum as DVT`: $||f - \frac{1}{n}\sum_i^nf_i||^2_2$ so $f^* = \frac{1}{n}\sum_i^nf_i$
>
> Due to this, our unweighted mean yields the same optimum as DVT. We will highlight this proof in the upcoming revision.
>
> > **Q5) Evaluation of non-salient objects**
>
> We agree that the main paper could benefit from additional visualizations. To clarify, `Figure 4` on page 5 of the main paper provides examples of open-vocabulary queries using the CLIP text branch for ***different objects within the same image***.
>
> We indeed evaluate our model on COCO benchmarks in `Table 1` on Page 5 of the main paper (COCO-Objects is listed as `Object`, COCO-Stuff is listed as `Stuff`). We also evaluate on other widely used segmentation benchmarks like PASCAL VOC, PASCAL Context, CityScape, and ADE20k. We reproduce the results for COCO here (mIoU, higher is better), and will update the naming in an upcoming revision:
>
> | Dataset     | CLIP | MaskCLIP | SCLIP | ClearCLIP | NACLIP | MaskCLIP + DVT | NACLIP + DVT | Ours (PH-Reg) |
> |-------------|:----:|:--------:|:-----:|:---------:|:------:|:--------------:|:------------:|:-------------:|
> | COCO Object | 6.50 |   26.94  | 33.52 |   32.77   |  33.15 |      20.89     |     32.89    |   **35.27**   |
> | COCO Stuff  | 7.19 |   18.8   | 22.65 |   23.89   |  23.30 |      17.10     |     23.41    |   **24.66**   |
>
> Additional segmentation visualization results are provided in `Figure 1` to `Figure 3` of the **supplemental PDF**, with heatmap visualizations of multiple objects within the same image provided from `Figure 4` to `Figure 7` of the supplemental.
>
> Our results demonstrate PH-Reg yields significant qualitative improvements in non-salient object localization.
>
> We appreciate this suggestion. We will update the revision to include additional visualizations in the main paper.
>
> > **Q6) DINOv2 performance compared to DVT**
>
> We would clarify that our PH-Reg method applied to DINOv2 ***obtains consistent performance improvements*** compared the unmodified DINOv2 as shown in `Table 2` on page 7 of the main paper. We also outperform the DVT denoiser on depth prediction tasks. Our method also has the advantage of not utilizing gradient-based neural field learning as done in DVT, offering ***significant training time and space usage*** improvements compared to DVT.
>
> Here we provide training time evaluations on a single GPU for DVT (***single GPU*** is specified in DVT's official code for all steps of feature extraction and neural field learning) and for fairness we also use a ***single GPU*** for our method's training. We follow all their official settings and use identical resolution and measure training time for the same ViT model.
>
> | Method | Stage 1 Extraction | Stage 1 Distillation | Stage 2 Training  | Total               |
> |--------|--------------------|----------------------|-------------------|---------------------|
> | DVT    | 2998 min           | 18340 min            |  570 min            | 21908 min (365.1 h) |
> | PH-REg | -                  | -                    |        -           | 9000 min (150 h)    |
>
> In practice, we parallelize training across 4 GPUs, and complete PH-Reg in roughly 1.5 days.
>
> A further advantage is the ***space utilization*** during training. DVT requires saving 1.4 terabytes of intermediate neural fields in the form of serialized Instant-NGP files. Our method computes all distillation targets on the fly, and requires no additional space.
>
> In summary, our method is significantly faster and space efficient compared to DVT -- while holding an advantage in depth prediction. We agree that future work should investigate how to further boost the performance in semantic segmentation when using PH-Reg.
>
>
> > **Conclusion**
>
> We thank the reviewer for the suggestions, and have added additional quantitative ablations over the shifting ratio and number of registers. Following your suggestion, we have also highlighted the optimality proof we previously included in the supplemental. We will also include additional visualizations in the revision showing the localization quality of different objects within the same image, and clarify the datasets that we evaluate on.
>
> We're grateful for your clear and helpful comments. In light of our response and the positive evaluation from other reviewers, ***we hope you might view our work in a more favorable light***. Please feel free to let us know if you have additional questions!

---

> > ### Comment · Area_Chair_kXcW · 2025-08-08
> > **NeurIPS Author–Reviewer Discussion Deadline Tomorrow**
> >
> > Dear Reviewers,
> >
> > This is a gentle reminder that the author–reviewer discussion period ends tomorrow. Constructive exchanges at this stage are critical to ensuring that all relevant clarifications and rebuttals are considered.
> >
> > Please also finalize your ratings and comments after the discussion period concludes, reflecting any changes in your assessment. Your timely participation will help ensure a fair and thorough review process.
> >
> > AC

---

### Note · Authors · 2025-08-12

We are grateful to all reviewers for their constructive suggestions, which we agree will significantly improve the communication of our work.

We are very encouraged by reviewers’ evaluation on the quality of this paper. All four reviewers find the work interesting ("**demonstrates consistent improvements across diverse dense prediction tasks**" (H7Es); "**an elegant solution to the problem**" (cHXN); "**presents a strong idea that significantly reduces the training costs**" (MibD); "**well-founded and addresses a relevant problem**" (EcZk)).


### General clarifications
### 1. Contributions

* We propose a test-time augmentation scheme that can effectively denoise dense features in vision transformers, **without the need for costly neural fields or gradient based optimization** (Figure 3).
* We elucidate the underlying components in a student model that contribute to learning **a clean dense feature map with minimal additional parameters** (Figure 2).
* We demonstrate that our method improves dense features in ViTs, leading to quantifiable improvements on downstream tasks that rely on fine-grained spatial understanding (e.g., semantic segmentation or depth prediction).
* We believe that the novel method introduced in this work will significantly impact how future researchers address artifacts in ViTs.

### 2. Additional experiments
* **[Additional ablations]** To address the suggestions of reviewers H7Es and cHXN, we add results that highlight the contribution of shifting ratio in our test time augmentation method, as well as the impact of the number of registers across different backbones. For reviewer EcZk, we include ablation studies on distillation approach and on different loss functions used during disillation.
* **[Additional comparisons]** For reviewers H7Es, cHXN and and EcZk, we add several results demonstrating the efficiency of PH-Reg compared to DVT. Additionally, for reviewer EcZk, we include a result comparing our method with the original register-ViT trained from scratch.
### 3. Writing
We thank all reviewers for suggestions regarding our writing and clarity. We have corrected some typos in our notation, and added theoretical details along with an optimality proof for the test-time augmentation.

### 4. Conclusion
We genuinely appreciate the suggestions, and believe our paper will be improved with your feedback! The additional experiments and clarifications will be reflected in the final version as well.


Best,

Authors

---

### Decision · Program_Chairs · 2025-09-17

**Decision:**

Accept (spotlight)

**Comment:**

This paper introduces PH-Reg, a post-hoc self-distillation framework designed to mitigate artifact tokens in pre-trained ViTs without requiring full retraining or labeled data. By leveraging frozen teachers, lightweight student updates, and augmentation-driven denoising, the approach enables efficient feature correction while maintaining generality.

All reviewers agree that the paper addresses a timely and relevant problem, providing a practical and efficient solution for enhancing pre-trained ViTs without costly retraining. The authors are highly encouraged to compare their method with the reference [52], particularly in the semantic segmentation setting, to better contextualize the contribution. Although [52] only treats high-norm tokens as defects, it similarly avoids expensive retraining, making a direct comparison valuable for deeper insights.

Overall, reviewers are generally positive, with most leaning toward acceptance after rebuttal clarifications.